# Anti-Inertia Disturbance Control of Permanent Magnet Synchronous Motor Based on Integral Time-Varying Fast Terminal Sliding Mode

**Fang Xie** [1,2,*], **Shilin Ni** [1,2], **Houying Wang** [1,2], **Jinghu Xu** [1,3] and **Ziang Zheng** [1,3]

1   School of Electrical Engineering and Automation, Anhui University, Hefei 230601, China;
z21301114@stu.ahu.edu.cn (S.N.); z21301196@stu.ahu.edu.cn (H.W.); z22301074@stu.ahu.edu.cn (J.X.);
z22301215@stu.ahu.edu.cn (Z.Z.)
2   National Engineering Laboratory of Energy-Saving Motor & Control Technology, Anhui University,
Hefei 230601, China
3   Engineering Research Center of Power Quality, Ministry of Education, Anhui University, Hefei 230601, China
*   Correspondence: xiefang@ahu.edu.cn

**Abstract:** To improve the speed control performance of a permanent magnet synchronous motor (PMSM) when the moment of inertia changes during operation, this paper studies an anti-inertia disturbance method based on integral time-varying fast terminal sliding mode control (ITFTSMC). First, the mechanical motion equation of the permanent magnet synchronous motor is established to obtain the inverse relationship between the moment of inertia and the rate of speed change. Second, the extended state observer is designed to identify the moment of inertia online and improve real-time tracking accuracy. Third, to solve the motor speed fluctuation caused by the sudden inertia change, an integral time-varying fast terminal sliding mode control method is proposed, which improves both speed stability through integral time-varying and rapidity through the fast terminal. The identified moment of inertia is then updated to the integral time-varying fast terminal sliding mode controller in real time to meet the dynamic performance of the permanent magnet synchronous motor when said inertia changes. Finally, simulation and comparative experiments were used to verify the feasibility and effectiveness of the abovementioned proposed methods.

**Keywords:** permanent magnet synchronous motor; moment of inertia identification; integral time-varying fast terminal sliding mode control

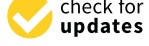



## 1. Introduction

Permanent magnet synchronous motors (PMSM) are widely used in various industrial fields due to their small size, high efficiency, convenient control, and ample speed regulation. In actual operations, however, its moment of inertia often changes, making the motor control parameters set before the servo system run out of sync with the controlled object during system operation. This leads to a deterioration in both the stability and rapidity of the speed of the servo control system. To maintain a fast and stable response to the system speed, the servo control system must obtain the information on inertia in real time and accordingly adjust the motor control parameters following the inertia parameter information to adapt to the changes in the moment of inertia.

Currently, numerous researchers have proposed equally numerous methods in anti-inertia disturbance, such as PI control [1–4], adaptive control [5–8], intelligent control [9–12], fuzzy control [13–15], predictive control [16–18], and sliding mode control [19–24], among others. Their methods are mainly divided into two categories: added observers and no observers.

There exist numerous studies on the anti-inertia disturbance of adding observers, all of which forward varying methods that effectively improve real-time tracking accuracy.

Most studies in the literature [1,2] generally use the gradient algorithm to identify the moment of inertia online and make real-time adjustments of the PI parameters of the velocity ring according to the principle of pole configuration. Although the effectiveness of the method can be verified by experiments, simple PI parameter adjustments cannot meet the complex parameter changes of multivariate systems. Other studies [3,4] also identify and compensate for the disturbance inertia using the inertia disturbance observer established by the Kalman filter. Said compensation system has good anti-inertia disturbance performance and robustness, but its adjustment of PI parameters still cannot meet the complex parameter change requirements. Some studies [5,6] follow the traditional disturbance observer as improved by the variable gain algorithm and are combined with the model reference adaptive algorithm. By then, the identified moment of inertia is updated for the improved disturbance observer, thus effectively improving the system's anti-disturbance performance. Other studies [7,8] use the forgetting factor recursive least squares algorithm to observe the moment of inertia and subsequently design the moment of inertia's adaptive controller to eliminate the influence of identification error on speed performance. There are also studies [5–8] showing good anti-inertia disturbance performance, but all have high hardware requirements and are difficult to implement in engineering.

There are also many studies on anti-inertia disturbances without adding observers; such methods make up for the lack of tracking accuracy by selecting control strategies. Some [9,10] propose an anti-inertia velocity control method for adaptive radial basis function networks that achieves high speed tracking accuracy (without current measurement or control) by directly acting on the voltage of the machine. Others [11,12] apply neural networks to PMSM control systems with inertia disturbances, simplify the control structure and parameter dependence, and ultimately achieve high servo accuracy under parameter changes. There also exists [13–15] various designs wherein a fuzzy inference module was used to apply to the adaptive current error integral-backward-step controller, thereby solving the problem of the influence of inertia disturbance on the system and improving the dynamic response performance of the system. Improved predictive control has also been applied to the PMSM current loop [16–18], effectively reducing the sensitivity of the servo system to motor inductance parameters and improving the high dynamic performance of the system. A non-singular fast terminal sliding mode control method has also been adopted [19] to suppress the influence of moment of inertia, thus ensuring that the speed tracking error converges quickly in a limited time and effectively weakens the speed fluctuation. Global fast terminal sliding mode control was also used [20] to quickly converge the system state to the equilibrium point, improve the response speed of the system, and enhance the robustness of the system. The note [21] proposes a robust integral sliding mode (RISM) manifold and a corresponding design method for stabilization control for uncertain systems with control input time delays. An improved sliding mode control, which is delay-dependent and suitable for small input time delays, keeps the system in the neighborhood of the RISM surface in finite time, improving the system's stability. The authors of [22] studied fault-tolerant control (FTC) designs based on nonsingular terminal sliding-mode control and nonsingular fast terminal sliding-mode control (NFTSMC). The proposed active FTC laws are shown to be able to achieve fault-tolerant objectives and maintain stabilization performance even when some of the actuators fail to operate. In order to achieve robust tracking performance, a novel sliding-mode control law [23] is designed in the SMC unit, and the adaptive unit is put forward to deal with uncertain gains. The article [24] proposes a continuous nonsingular terminal sliding-mode control with integral-type sliding surfaces (CNTSMC-ISS) framework for disturbed systems. Compared with the existing sliding-mode controllers, the noteworthy contributions are the alleviation of the chattering phenomenon, the fast finite-time stability, and the singularity-free and ease-of-implementation characteristics. These methods have also achieved satisfactory results in the experiment, although the lack of real-time observation of inertia cannot meet the requirements of the system for high tracking accuracy.

In order to improve the tracking accuracy of the system to the motor speed, this paper improves the extended state observer to a new inertia observer. By adding a time-varying function, the gain increases slowly, reducing the impact on the motor speed while observing the inertia. To improve the stability and rapidity of the motor speed when the inertia changes, this paper proposes an integral time-varying fast terminal sliding mode control (ITFTSMC) method. In order to shorten the duration of the approaching motion stage and ensure the dynamic quality of the approaching motion, the author improves the exponential approaching law and proposes a new type of approaching law. In order to improve the stability and fast performance of the motor speed, the author designed a new type of integral time-varying module and fast terminal module to improve the conventional sliding mode controller. The integral time-varying module can improve the stability of the motor speed and keep the speed overshoot within a small error range, and the fast terminal module can improve the speed performance of the motor speed so that the speed can recover to stability in a short time. Finally, experiments were designed to verify the effectiveness of the method.

## 2. Mathematical Model of the Moment of Inertia of PMSM

To solve the influence of the disturbance of the moment of inertia on the speed of the motor operation, the current study starts with the mechanical motion equation of the permanent magnet synchronous motor in the *d-q* coordinate system. It also obtains the correlation between the moment of inertia and the motor speed through the analysis of this equation.

The mechanical equation of motion of PMSM in the *d-q* coordinate system is shown in Formula (1) below:

$$J\frac{d\omega_m}{dt} = T_e - T_L - B\omega_m \tag{1}$$

where $J$ is the moment of inertia; $\omega_m$ is the mechanical angular velocity; $T_e$ is the electromagnetic torque; $T_L$ is load torque; $B$ is the damping coefficient.

In the moment of inertia identification algorithm, the sampling frequency of the actual speed of the motor is relatively high; hence, the damping coefficient of the system can be ignored. Therefore, the above formula can be simplified into Formula (2) below:

$$\frac{d\omega_m}{dt} = \frac{T_e - T_L}{J} \tag{2}$$

The logical relationship between motor speed and mechanical angular velocity is shown in Equation (3) below:

$$n = \frac{30}{\pi} \cdot \omega_m \tag{3}$$

where $n$ is the motor speed.

From Equations (2) and (3), the moment of inertia is clearly inversely proportional to the rate of change of the mechanical angular velocity, and the mechanical angular velocity is proportional to the rotational speed; hence, the rate of change of the moment of inertia and the speed are also inversely proportional. As shown in Equation (4):

$$\frac{dn}{dt} = \frac{\pi}{30} \cdot \frac{T_e - T_L}{J} \tag{4}$$

Obviously, when the load torque remains constant, the moment of inertia becomes a disturbance factor that affects the changes in motor speed. In order to quickly restore stability to the speed, this article identifies the moment of inertia and updates the identification value in real-time to the speed loop, which improves the fast performance and stability of the speed.

### 3. Inertia Identification Based on Extended State Observer

For the speed to quickly return to a stable state when the moment of inertia changes, accurate and rapid real-time monitoring of the inertia change is required. In this paper, an extended state observer (ESO) is used to identify the inertia online.

The motor angular velocity expression shown in Equation (2) is transformed into:

$$\frac{d\omega_m}{dt} = \frac{1}{J}\left(K_t i_q - T_L\right) = \frac{1}{J}(K_t i_q^* - T_L) + \frac{K_t}{J}(i_q - i_q^*) = \frac{1}{J}(K_t i_q^* - T_L) + L_\omega, \quad (5)$$

in the formula, $K_t$ is the torque constant, $K_t = \frac{3}{2}p_n \cdot \psi_f$; $p_n$ is the polar logarithm; $\psi_f$ is the permanent magnet flux chain; $J$ is the reference value of inertia; $i_q^*$ is the reference current; $i_q$ is the actual current; and $L_\omega$ is the disturbance of the speed loop.

For the motor angular velocity equation shown in Equation (5), the extended state observer is designed as follows:

$$\begin{cases} e = \omega_m^* - \omega_m \\ \dot{\omega}_m = \frac{1}{J}(K_t i_q^* - T_L) + L_\omega - \beta_1 e, \\ L_\omega = -fal(e, \lambda, \delta) \end{cases} \quad (6)$$

in the equation, $\omega_m^*$ is the motor reference mechanical angular velocity, is usually a constant, and $\omega_m$ is the actual mechanical angular velocity; $\beta_1$, $\alpha$, $\lambda$ and $\delta$ are the parameters of the observer, where $\beta_1 = 0.1$, $\lambda = 0.8$, and $\delta = 0.01$. The function $fal$ is represented as:

$$fal(e, \lambda, \delta) = \begin{cases} |e|^\lambda \mathrm{sgn}(e), & |e| > \delta \\ \frac{e}{\delta^{1-\lambda}}, & |e| < \delta \end{cases}, \quad (7)$$

where $\delta$ is the interval length of the linear segment; $\lambda$ is a nonlinear factor; and sgn is a sign function.

In the observer, the high gain of $\beta_1$ can improve the accuracy and bandwidth of the observer. However, when the initial value of the observer is inconsistent with the actual speed of the motor, the high gain of the observer can cause the speed to overshoot, which has a negative impact on speed control. To reduce the impact of speed overshoot, this article utilizes the saturation characteristics of the sgn function and replaces $\beta_1$ with a time-varying function $\beta_{11}$, so that the observer adopts a smaller gain during operation. As time increases, the observer's gain gradually increases until the time-varying function saturates. The expression of the extended state observer based on time-varying parameters is:

$$\begin{cases} e = \omega_m^* - \omega_m \\ \dot{\omega}_m = \frac{1}{J}\left(K_t i_q^* - T_L\right) + L_\omega - \beta_{11} e \\ L_\omega = -fal(e, \lambda, \delta) \\ \beta_{11} = \begin{cases} b_1 t & , t \le t_s \\ b_2 sign(\eta t), & t > t_s \end{cases} \end{cases}, \quad (8)$$

in the formula, $b_1$ and $b_2$ are both gain coefficients, where $b_1 = 0.1$, $b_2 = 0.12$, and $t_s$ is the set time constant.

This article sets the load torque to remain constant with only changes in inertia, so the speed loop disturbance is caused by changes in inertia. As shown in the following equation, the rate of change of $\omega_m$ can also be written as:

$$\dot{\omega}_m = \frac{1}{\hat{J}}(K_t i_q^* - T_L) \quad (9)$$

where $\hat{J}$ is the inertia identification value. Equation (10) can be obtained by combining Equations (8) and (9).

$$\frac{1}{\hat{J}} = \frac{1}{J} + \frac{L_\omega - \beta_{11}e}{K_t i_q^* - T_L} \tag{10}$$

Figure 1 shows the simulation diagram of inertia identification.

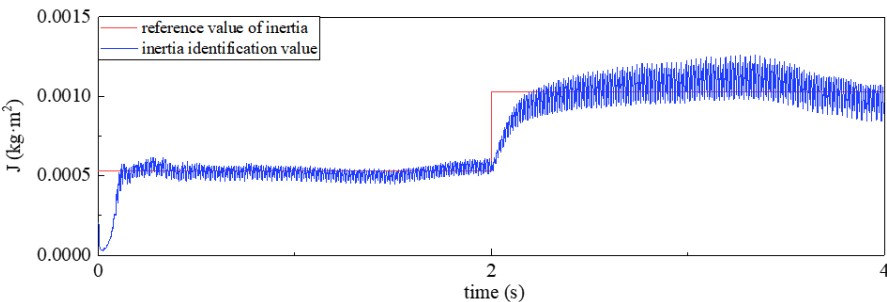

**Figure 1.** Identification result of inertia.

As shown in the figure, good observation results can be obtained by using the extended state observer to observe the moment of inertia.

## 4. Speed Controller Design

To improve the dynamic performance of the speed at the time of inertia change, an integral time-varying fast terminal sliding mode control (ITFTSMC) is designed to be applied to the speed loop. Figure 2 shows the system control block diagram.

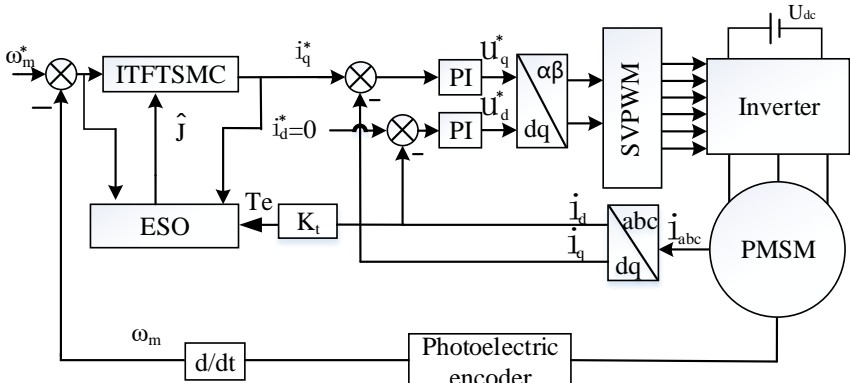

**Figure 2.** System control block diagram.

The input of the system was the speed error; the output was the given value of the current; the speed signal of PMSM was fed back through the optical encoder; the current loop adopted PI control; the q-axis current was the velocity ring output value; and the d-axis current was taken as 0. The adjustment and update of ITFTSMC control law parameters was carried out by feeding back to the inertia recognition value of the speed controller, which was derived from the online identification of the improved Inertia observer. The system modulation method adopts the spatial vector pulse width modulation (SVPWM) scheme.

### 4.1. ITFTSMC Analytical Model

Define the status variables of the PMSM system as:

$$\begin{cases} x_1 = \omega_m^* - \omega_m \\ x_2 = \dot{x}_1 = -\dot{\omega}_m \end{cases} . \tag{11}$$

This is obtained according to Equations (5) and (11):

$$\begin{cases} \dot{x}_1 = -\dot{\omega}_m = \frac{1}{J}\left(T_L - K_t i_q\right) \\ \dot{x}_2 = -\ddot{\omega}_m = -\frac{K_t}{J}\dot{i}_q \end{cases} \tag{12}$$

In order to improve the fast and stable performance of the rotating speed of the motor, a new sliding mode surface is defined in this paper by combining the integral time-varying module and the fast terminal module. The function of the sliding mode surface is shown in Equation (13):

$$s = x_1 + c\int_0^t x_1 d\tau + \alpha e^{-\beta \cdot t} + \rho|x_1|^{p/q}\text{sgn}(x_1) \tag{13}$$

where $c, \alpha, \beta, \rho, p, q$ are constants, $p, q$ are Positive odd numbers, and, $c = 50$, $\beta = 100$, $\rho = 50$, and $p/q = 3/2$.

The derivation of Formula (13) can be obtained:

$$\dot{s} = \dot{x}_1 + cx_1 - \alpha\beta e^{-\beta t} + \rho\frac{p}{q}|x_1|^{\frac{p}{q}-1}\dot{x}_1\text{sgn}(x_1) \tag{14}$$

To achieve the global robustness of sliding mode control, the system trajectory needs to be on the time-varying sliding mode surface at the initial moment, so that $s = 0$ yields:

$$\alpha = -x_1(0) - c\int_0^t x_1(0)d\tau - \rho|x_1(0)|^{p/q}\text{sgn}(x_1(0)). \tag{15}$$

In order to shorten the time of approaching motion and improve the dynamic quality of approaching motion, this paper improves the exponential approach law and designs a new approach law, as shown in Equation (16):

$$\dot{s} = -k_1|s|^\partial\text{sgn}(s) - k_2 s \tag{16}$$

where $\partial = 1/2$, $k_1 = 200$, $k_2 = 300$.

From Equations (12), (14), and (16), it can be obtained that the actual current of the q-axis is as follows:

$$i_q = \frac{J}{K_t}\left[\frac{1}{1+\rho\frac{p}{q}|x_1|^{\frac{p}{q}-1}\text{sgn}(x_1)}\right]\left[k_1|s|^\partial\text{sgn}(s) + k_2 s + cx_1 - \alpha\beta e^{-\beta \cdot t}\right] + \frac{T_L}{K_t}. \tag{17}$$

By substituting the rotational inertia identification value $\hat{J}$ into Equation (17), the reference current of the q-axis can be obtained as follows:

$$i_q^* = \frac{\hat{J}}{K_t}\left[\frac{1}{1+\rho\frac{p}{q}|x_1|^{\frac{p}{q}-1}\text{sgn}(x_1)}\right]\left[k_1|s|^\partial\text{sgn}(s) + k_2 s + cx_1 - \alpha\beta e^{-\beta \cdot t}\right] + \frac{T_L}{K_t}. \tag{18}$$

*4.2. ITFTSMC Submodule Analysis*

To improve the dynamic performance of the speed, this paper improves the traditional sliding mode controller and adds both the integral module, the time-varying module, and the fast terminal module one by one based on this. The sliding surface of the conventional sliding mode controller is $s = x_1 + cx_2$, the sliding surface of the integral sliding mode controller is $s = x_1 + c\int_0^t x_1 d\tau$, the sliding surface of the integral time-varying sliding mode controller is $s = x_1 + c\int_0^t x_1 d\tau + \alpha e^{-\beta \cdot t}$, and the sliding surface of the integral time-varying fast terminal sliding mode controller is $s = x_1 + c\int_0^t x_1 d\tau + \alpha e^{-\beta \cdot t} + \rho|x_1|^{p/q}\text{sgn}(x_1)$. The reaching laws of the four sliding mode controllers are $\dot{s} = -k_1|s|^\partial\text{sgn}(s) - k_2 s$.

Figure 3 shows the rotational speed comparison waveform brought about by conventional sliding mode control and integral sliding mode control (ISMC).

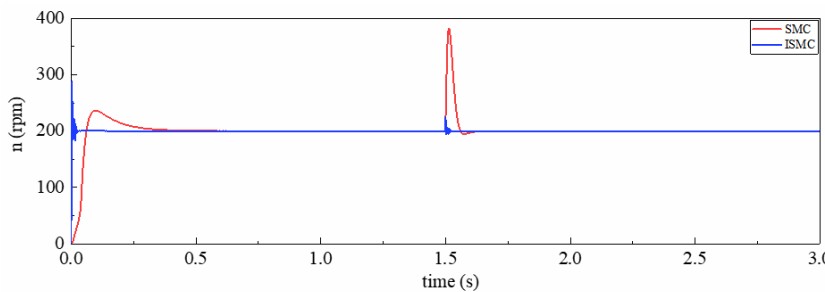

**Figure 3.** Speed contrast waveform of SMC and ISMC.

Figure 4 shows the comparison waveform of the q-axis current caused by conventional sliding mode control and integral sliding mode control.

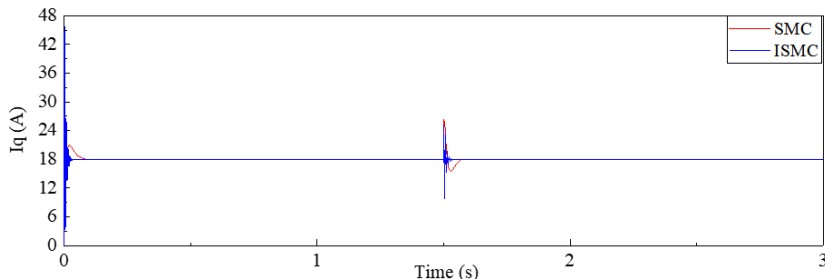

**Figure 4.** Iq contrast waveform of SMC and ISMC.

As shown in Figure 3, the speed of SMC has a large overshoot and long response time when the motor starts and the inertia changes, and the speed waveform obtained by the sliding mode controller after adding the integral module has a great improvement in response time. However, the overshoot remains large and needs further improvement. As shown in Figure 4, the fluctuation trend of the q-axis current and the fluctuation trend of the speed are roughly the same, but the current fluctuation amplitude of ISMC is extremely large, which is not conducive to the operation of the motor and needs further improvement.

To reduce the high overshoot, the current study adds a time-varying module based on ISMC to obtain integrated time-varying sliding mode control (ITSMC). Figure 5 shows the speed comparison waveform under SMC, ISMC, and ITSMC.

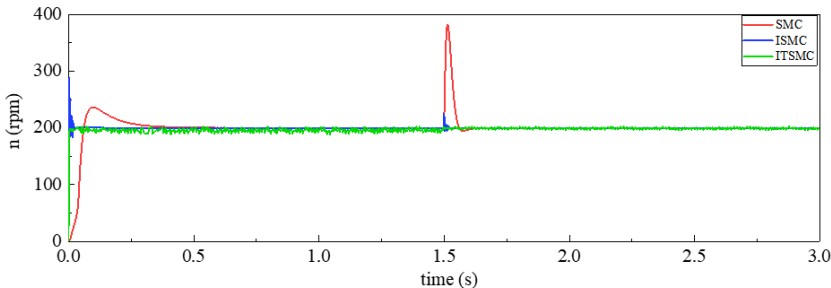

**Figure 5.** Speed contrast waveforms of SMC, ISMC, and ITSMC.

Figure 6 shows the q-axis current comparison waveform under SMC, ISMC, and ITSMC.

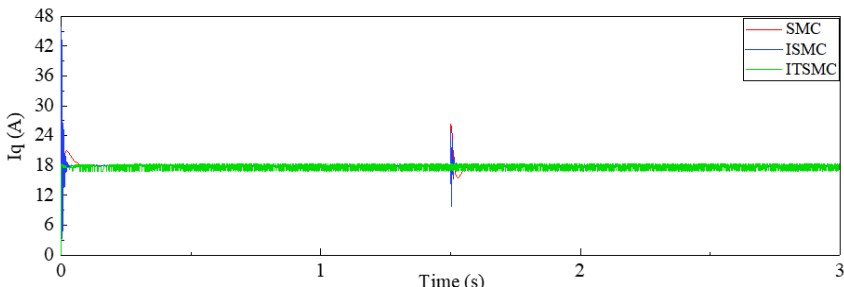

**Figure 6.** Iq contrast waveform of SMC, ISMC, and ITSMC.

Following Figure 5, the speed of ISMC has a large overshoot when the motor is started, and the speed waveform obtained by the sliding mode controller after adding the time-varying module has greatly improved the overshoot. As shown in Figure 6, the overall current fluctuation of ITSMC slightly increases, but the overshoot is much smaller than that of ISMC, and there is still room for improvement.

To reduce the large response time, the fast terminal module was added to the ITSMC to obtain the integrated time-varying sliding mode control (ITFTSMC). Figure 7 shows the speed comparison waveform under SMC, ISMC, ITSMC, ITFTSMC, and ITFTSMC.

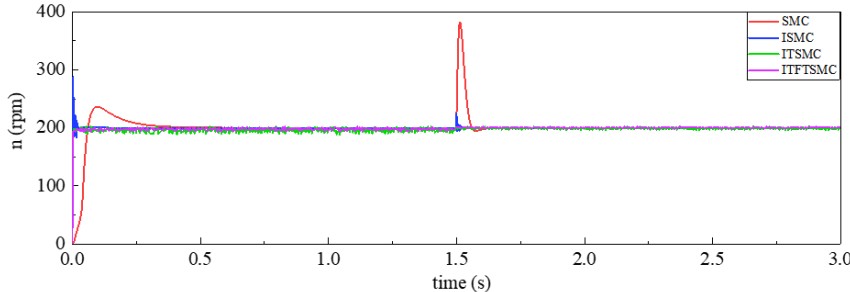

**Figure 7.** Speed contrast waveforms of SMC, ISMC, ITSMC, and ITFTSMC.

Figure 8 shows the q-axis current comparison waveform under SMC, ISMC, ITSMC, and ITFTSMC.

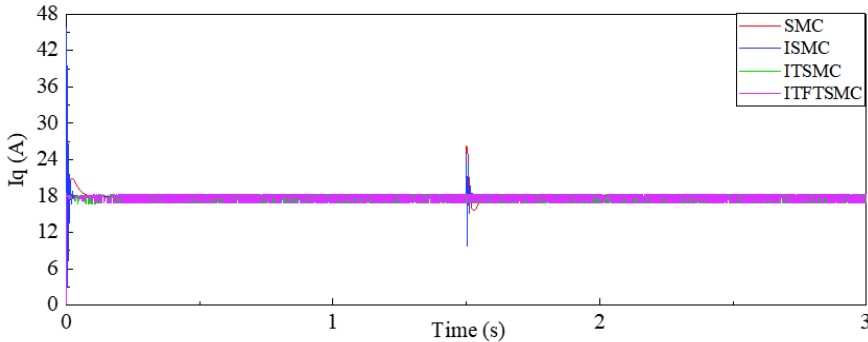

**Figure 8.** Iq contrast waveforms of SMC, ISMC, ITSMC, and ITFTSMC.

As shown in Figure 7, after the introduction of the fast terminal module in ITSMC, the response time of the motor during startup is significantly reduced, and there is almost no overshoot. Compared to the other three sliding mode controllers, ITFTSMC provides a faster and more stable response speed during motor startup. When the moment of inertia changes, both SMC and ISMC have larger overshoots and longer response times, while both ITSMC and ITFTSMC have smaller response times and almost no overshoots. Overall, when the moment of inertia suddenly changes, the rotational speed of the ITFTSMC has the fastest and most stable performance. Additionally, as shown in Figure 8, the q-axis current

of ITFTSMC also has the same characteristics, with the best response time and overshoot, which meet the design requirements.

### 4.3. Stability Analysis

To verify the stability of the integral time-varying fast terminal sliding mode controller, the Lyapunov function was selected, which is expressed as: $V = 1/2s^2$.

The derivation of this function yields $\dot{V} = s\dot{s}$.

Substituting Equations (12) and (14) into the above equation, the following equation is obtained:

$$\begin{aligned}\dot{V} = s\dot{s} &= s\left(\dot{x}_1 + cx_1 - \alpha\beta e^{-\beta t} + \rho\frac{p}{q}|x_1|^{\frac{p}{q}-1}\cdot\dot{x}_1\text{sgn}(x_1)\right)\\&= s\left[\left(1 + \rho\frac{p}{q}|x_1|^{\frac{p}{q}-1}\text{sgn}(x_1)\right)\left(-\frac{K_t}{J}i_q^* + \frac{T_L}{J}\right) + cx_1 - \alpha\beta\cdot e^{-\beta t}\right]\end{aligned} \qquad (19)$$

Substitute Equation (17) into Equation (19) to obtain:

$$\begin{aligned}\dot{V} &= s\cdot\left[\left(1 + \rho\frac{p}{q}|x_1|^{\frac{p}{q}-1}\text{sgn}(x_1)\right)\left(-\frac{K_t}{J}i_q^* + \frac{T_L}{J}\right) + cx_1 - \alpha\beta\cdot e^{-\beta t}\right]\\&= s\cdot\left[-k_1|s|^{\partial}\text{sgn}(s) - k_2s\right] \le -k_1|s|^{\partial+1} - s^2 < 0\end{aligned} \qquad (20)$$

Obviously, the sliding mode controller designed in this paper satisfies the Lyapunov stability theorem: $\dot{V} = s\dot{s} < 0$. The system is asymptotically stable.

## 5. Experimental Validation

### 5.1. Construction of the Experimental Platform

To verify the effectiveness of the integral-based time-dependent fast terminal sliding mode control method proposed in this paper, a PMSM control platform with tunable inertia is built as shown in Figure 9.

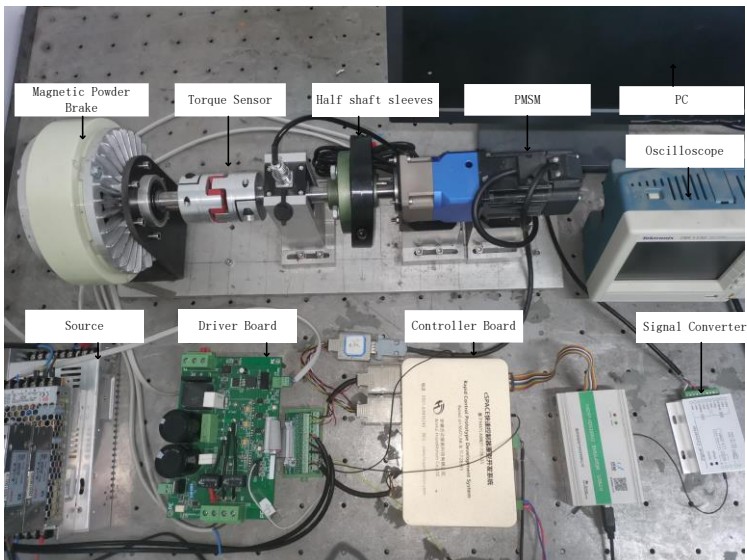

**Figure 9.** Experimental platform.

The platform is mainly comprised of eight parts: a permanent magnet synchronous motor; a magnetic powder clutch; a torque sensor; a half-shaft bushing; CSPACE; a 2500PPR incremental encoder; an oscilloscope; and a motor drive circuit board. Among these, the change in PMSM moment of inertia is realized by the increase and decrease of the half-shaft casing. Encoders are then used for the collection of motor speeds, and CSPACE is a

semi-physical simulation system based on TMS320F28335DSP. It has AD, DA, IO, Encoder, PWM, and other simulation functions. After the control algorithm has been designed in MATLAB/Simulink, the DSP code can be generated, and the corresponding control signals can be generated.

Table 1 shows the motor parameters of the experimental platform.

**Table 1.** PMSM parameters.

| Parameter | Value |
| --- | --- |
| Number of pole pairs | 4 |
| Line resistance | 0.33 Ω |
| Line inductance | 0.9 mH |
| Rated voltage | 36 V |
| Rated torque | 1.276 Nm |
| Rated current | 7.5 A |
| Maximum speed | 3000 rpm |
| Motor weight | 1.1 kg |
| Nominal power | 270 w |
| Nominal inertia | $6.3 \times 10^{-4}$ kg·m$^2$ |

Table 2 shows the key parameters of the inertia observer.

**Table 2.** Parameters of the inertia observer.

| Parameter | Value |
| --- | --- |
| interval length of the linear segment($\delta$) | 0.01 |
| nonlinear factor($\lambda$) | 0.8 |
| gain coefficient($b_1$) | 0.1 |
| gain coefficient($b_2$) | 0.12 |

Table 3 shows the key parameters of ITFTSMC.

**Table 3.** Parameters of the ITFTSMC.

| Parameter | Value |
| --- | --- |
| Integral coefficient($c$) | 50 |
| time-varying parameter($\beta$) | 100 |
| Fast terminal coefficient($\rho$) | 50 |
| Fast terminal coefficient($p/q$) | 3/2 |
| Sliding surface coefficient($a$) | 1/2 |
| Sliding surface coefficient($k_1$) | 200 |
| Sliding surface coefficient($k_2$) | 300 |

*5.2. Inertia Identification*

In order to better restrain the influence of inertia disturbance on motor speed, this paper uses the expanded state observer to observe the moment of inertia online. The following figure shows the experimental results of inertia identification.

As shown in Figure 10, the inertia can be identified using the extended state observer, which can have a faster identification speed and higher identification accuracy. Obviously, the extended state observer is suitable for the control system applied in this paper.

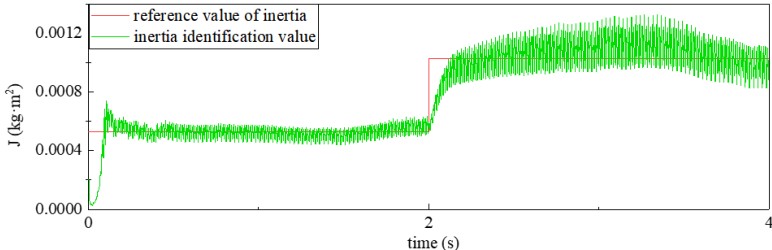

**Figure 10.** Identification result of inertia.

### 5.3. Motor Speed Performance Verification

5.3.1. Speed Performance Verification during Motor Start

To verify the influence of motor starting on speed performance, the change of inertia was ignored, the speed response waveforms under the four groups of methods were compared, and the moment of inertia was set to 0.0001 kg·m². Figure 11 shows the rotational speed waveforms of the four sets of methods under low, medium, and high-speed conditions.

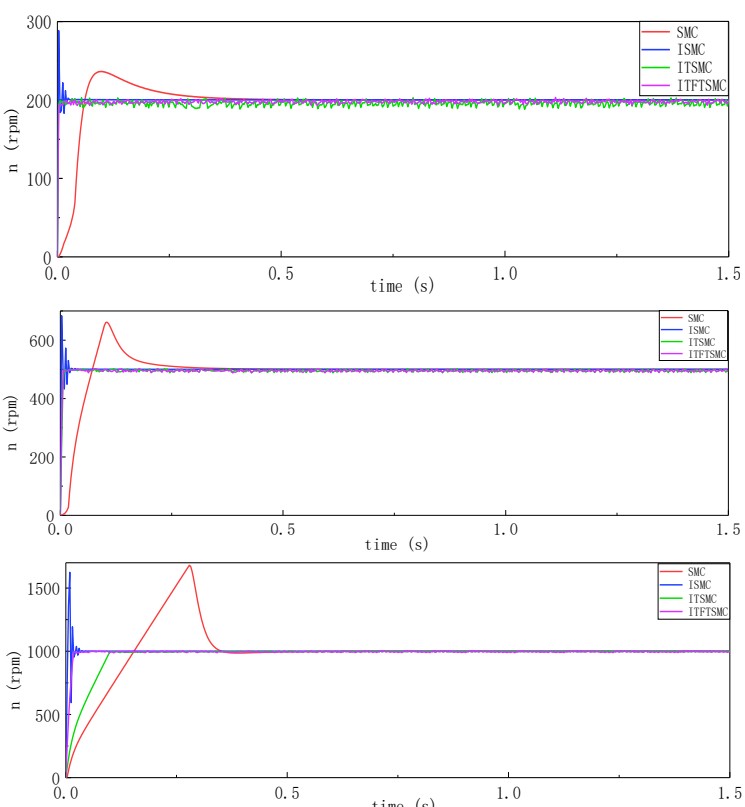

**Figure 11.** Speed response waveform when the motor starts.

For easier comparison and illustration, the overshoot and response time of different sliding mode controllers during motor startup under low, medium, and high speed conditions are provided to evaluate the effectiveness of the ITFTSMC method. The data comparison is shown in Table 4. Among them, SMC fluctuates the most: the speed overshoot at low, medium, and high speeds is more than 15% higher than ITFTSMC, and the response time is more than 0.3 s longer than ITFTSMC. The response time of TSMC is very short, although the overshoot is more than 30% higher than that of ITFTSMC. ITSMC generates more fluctuations at low speeds than ITFTSMC, and its response time at high speeds is about 75 ms longer than ITFTSMC. Obviously, the speed stability and fast performance of ITFTSMC during motor startup are superior to those of the other three controllers.

**Table 4.** Data comparison table during motor startup.

| Experiment | Controller | Overshoot | Adjust Time |
|---|---|---|---|
| Speed response waveform when the motor starts (low speed) | SMC | 16.5% | 407 ms |
| | ISMC | 42% | 43 ms |
| | ITSMC | 3.5% | 18 ms |
| | ITFTSMC | 2% | 16 ms |
| Speed response waveform when the motor starts (medium speed) | SMC | 32% | 326 ms |
| | ISMC | 37% | 48 ms |
| | ITSMC | 4.7% | 13 ms |
| | ITFTSMC | 4.2% | 12 ms |
| Speed response waveform when the motor starts (high speed) | SMC | 59% | 402 ms |
| | ISMC | 57% | 63 ms |
| | ITSMC | 5.6% | 113 ms |
| | ITFTSMC | 4.8% | 38 ms |

5.3.2. Influence of Inertia Disturbance on Static Rotational Speed Performance

Effect of Small Inertia Disturbance on Static Speed Performance

To verify the influence of small inertia disturbances on the static speed performance, keep the inertia at a given speed unchanged, and the moment of inertia increases from $10^{-4}$ kg·m$^2$ to $2 \times 10^{-4}$ kg·m$^2$ at 2 s, and the speed response waveform at this time is then compared. Figure 12 shows the rotational speed waveforms of the four sets of methods under low, medium, and high-speed conditions.

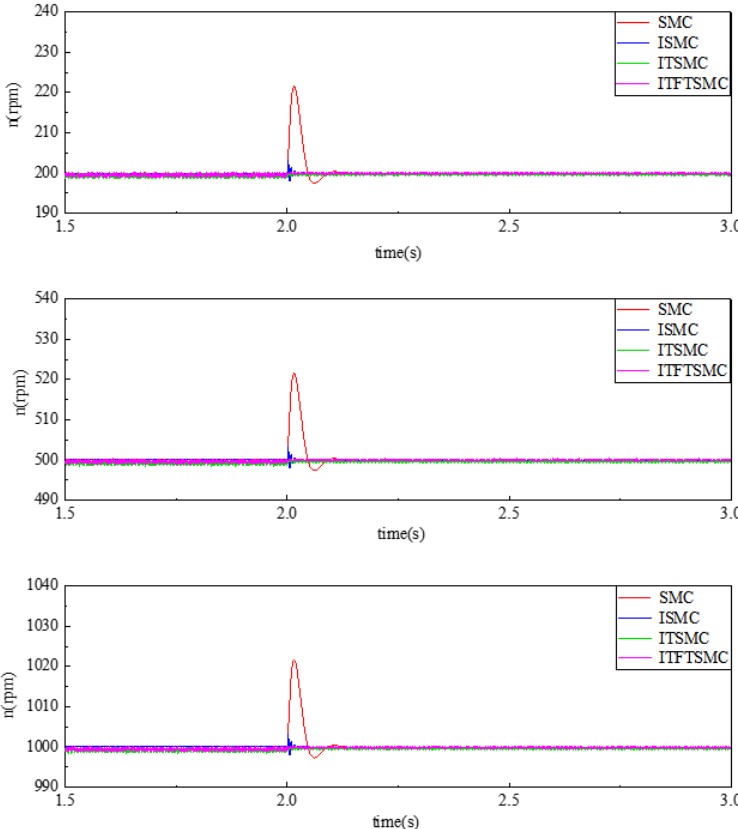

**Figure 12.** Static rotational speed response waveform under small inertia disturbance.

When the rotational inertia increases from $10^{-4}$ kg·m$^2$ to $2 \times 10^{-4}$ kg·m$^2$, the overshoot and response time under different sliding mode controllers are shown in Table 5. The rotational speed of a conventional SMC changes most significantly, with the highest overshoot and the longest response time; an ITFTSMC has the smallest speed fluctuation and the shortest response time. Under low, medium, and high speed conditions, the speed fluctuation of conventional SMC is about 20 rpm higher than that of ITFTSMC, and the response time is about 0.1 s slower; The speed fluctuation of TSMC is about 4 rpm higher than that of ITFTSMC, and the response time is about 20 ms slower; The rotational speed response time of ITSMC is similar to that of ITFTSMC, but the overall fluctuation of its rotational speed is greater than that of ITFTSMC. Experiments have proven that when the inertia changes in a small range, the rotational speed of ITFTSMC has the best stability and fast performance at both low, medium, and high speeds.

**Table 5.** Comparison table of static speed data under small inertia changes.

| Experiment | Controller | Overshoot | Adjust Time |
|---|---|---|---|
| Static rotational speed response waveform under small inertia disturbance (low speed) | SMC | 11% | 107 ms |
| | ISMC | 2% | 26 ms |
| | ITSMC | 0.34% | 8 ms |
| | ITFTSMC | 0.27% | 7 ms |
| Static rotational speed response waveform under small inertia disturbance (medium speed) | SMC | 4% | 112 ms |
| | ISMC | 0.8% | 27 ms |
| | ITSMC | 0.073% | 7.8 ms |
| | ITFTSMC | 0.058% | 6.9 ms |
| Static rotational speed response waveform under small inertia disturbance (high speed) | SMC | 2% | 124 ms |
| | ISMC | 0.47% | 27 ms |
| | ITSMC | 0.038% | 8 ms |
| | ITFTSMC | 0.029% | 7 ms |

Effect of Moderate Inertia Disturbance on Static Speed Performance

To verify the influence of moderate inertia disturbance on the static speed performance, keep the inertia at a given speed unchanged, and the moment of inertia increases from $10^{-4}$ kg·m$^2$ to $10^{-3}$ kg·m$^2$ at 2 s, and the speed response waveform at this time is then compared. Figure 13 shows the rotational speed waveforms of the four sets of methods under low, medium, and high-speed conditions.

When the moment of inertia increases from $10^{-4}$ kg·m$^2$ to $10^{-3}$ kg·m$^2$, the overshoot and response times under different sliding mode controllers are shown in Table 6. The SMC speed change remains the largest, and the ITFTSMC speed change remains the smallest. Under low, medium, and high speed conditions, SMC speed overshoot is more than 15% higher than ITFTSMC, and the response time is more than 110 ms longer than ITFTSMC; TSMC overshoot is more than 1.5% higher than ITFTSMC, and the response time is more than 25 ms longer than ITFTSMC; The rotational speed fluctuation caused by ITSMC before and after the inertia mutation is greater than that of ITFTSMC, and the rotational speed fluctuation at high speed is the most obvious. Experiments have shown that when inertia changes in a medium range, its rotational speed still has the best stability and fastest performance.

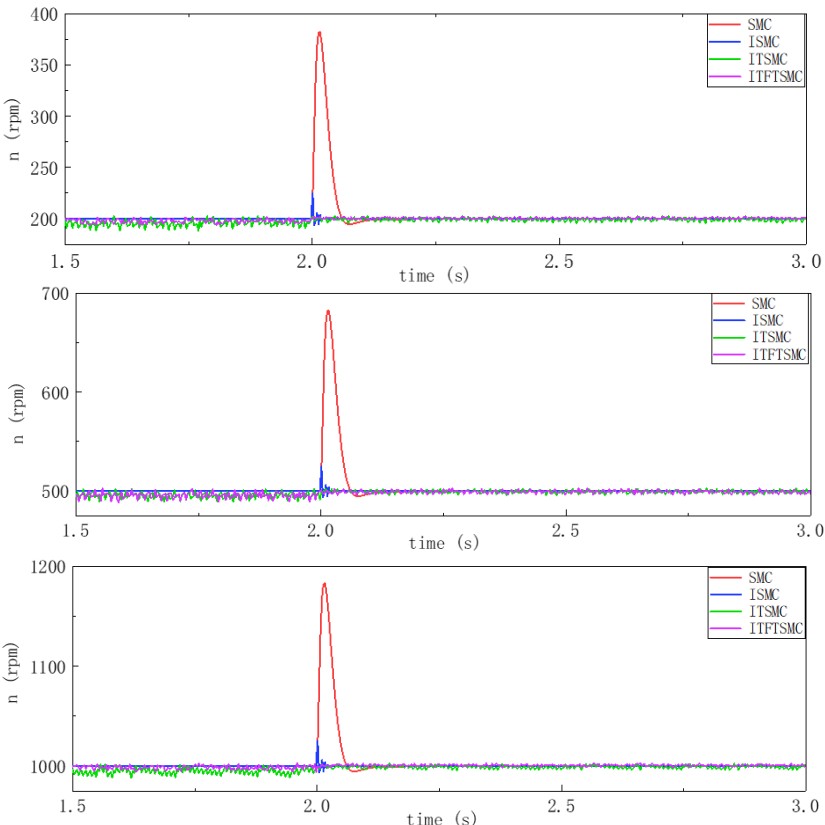

**Figure 13.** Static rotational speed response waveform under moderate inertia disturbance.

**Table 6.** Comparison table of static speed data under medium inertia change.

| Experiment | Controller | Overshoot | Adjust Time |
|---|---|---|---|
| Static rotational speed response waveform under moderate inertia disturbance (low speed) | SMC | 87.5% | 125 ms |
| | ISMC | 12.5% | 27 ms |
| | ITSMC | 4% | 1.2 ms |
| | ITFTSMC | 2.5% | 1.1 ms |
| Static rotational speed response waveform under moderate inertia disturbance (medium speed) | SMC | 35% | 120 ms |
| | ISMC | 5% | 26 ms |
| | ITSMC | 1.6% | 3.4 ms |
| | ITFTSMC | 1.5% | 2.9 ms |
| Static rotational speed response waveform under moderate inertia disturbance (high speed) | SMC | 17.5% | 118 ms |
| | ISMC | 2.5% | 27 ms |
| | ITSMC | 1.3% | 1.4 ms |
| | ITFTSMC | 0.89% | 0.83 ms |

Effect of Large Inertia Disturbance on Static Speed Performance

To verify the influence of large inertia disturbances on the static speed performance, the moment of inertia increases from $10^{-3}$ kg·m$^2$ to $10^{-2}$ kg·m$^2$ at 2 s to compare the speed response waveform at this time. Figure 14 shows the corresponding rotational speed waveform.

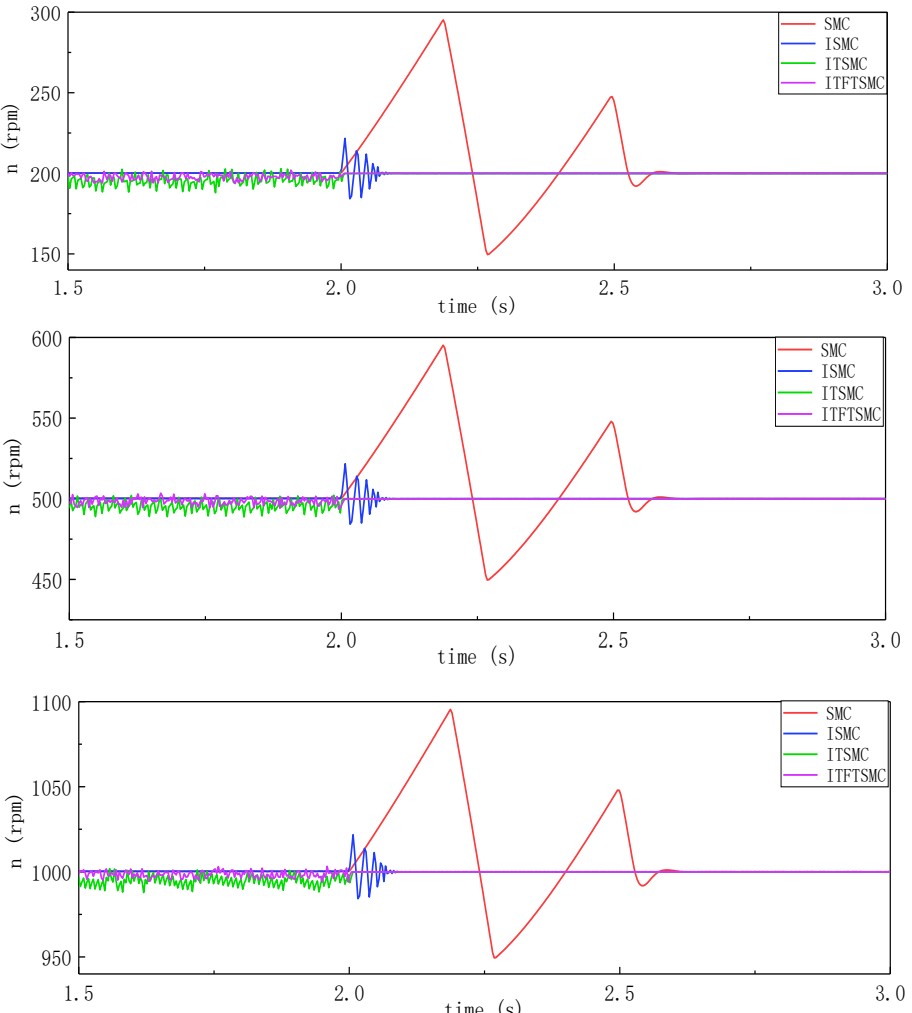

**Figure 14.** Static rotational speed response waveform under large inertia disturbance.

When the moment of inertia increases from $10^{-3}$ kg·m$^2$ to $10^{-2}$ kg·m$^2$, the overshoot and response time under different sliding mode controllers are shown in Table 7. The rotational speed of the SMC changes most acutely. Under low, medium, and high speed conditions, its rotational speed increases by about 90 rpm, and the response time also exceeds 0.5 s. The rotation speed fluctuation of ISMC is relatively small, but the rotation speed increases by about 20 rpm. The response time also reached about 0.1 s. The rotational speed fluctuations of ITSMC and ITFTSMC are minimal, but considering that the rotational speed fluctuations of ITSMC before the inertia change are greater than the rotational speed fluctuations of ITFTSMC, it can be concluded that when the inertia changes over a wide range, the rotational speed of ITFTSMC still has the best stability and fast performance.

Effect of Inertia Disturbance on Electric Current at Static Rotation Speed

To verify the effect of inertia perturbation on the current at static speed, this article sets the speed to 500 rpm, and the inertia increases from 2 s, respectively, from $10^{-4}$ kg·m$^2$ to $2 \times 10^{-4}$ kg·m$^2$, $10^{-4}$ kg·m$^2$ to $10^{-3}$ kg·m$^2$, and $10^{-3}$ kg·m$^2$ to $10^{-2}$ kg·m$^2$. Comparing the current response waveforms under these three inertia changes, Figure 15 shows the corresponding current waveforms.

**Table 7.** Comparison table of static speed data under large inertia changes.

| Experiment | Controller | Overshoot | Adjust Time |
|---|---|---|---|
| Static rotational speed response waveform under large inertia disturbance (low speed) | SMC | 45% | 560 ms |
| | ISMC | 9.8% | 123 ms |
| | ITSMC | 6% | 4.2 ms |
| | ITFTSMC | 4.3% | 3.7 ms |
| Static rotational speed response waveform under large inertia disturbance (medium speed) | SMC | 18% | 556 ms |
| | ISMC | 4% | 118 ms |
| | ITSMC | 2.4% | 3.8 ms |
| | ITFTSMC | 1.4% | 3.9 ms |
| Static rotational speed response waveform under large inertia disturbance (high speed) | SMC | 9% | 547 ms |
| | ISMC | 2% | 107 ms |
| | ITSMC | 1.3% | 4 ms |
| | ITFTSMC | 0.83% | 3.9 ms |

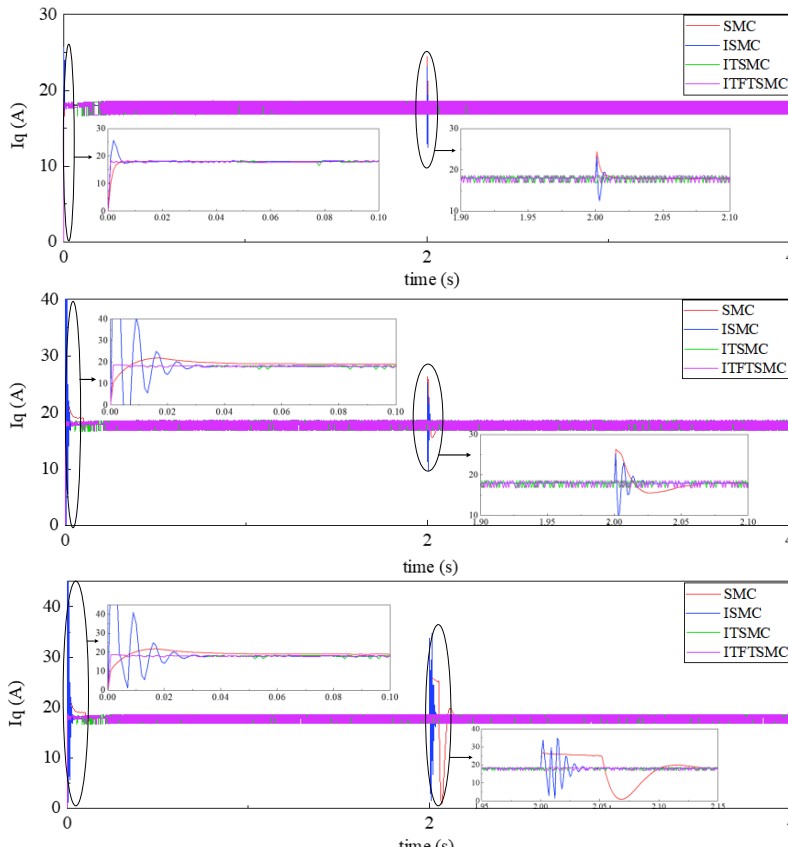

**Figure 15.** Current response waveform under inertia disturbance at static rotational speed.

The amplitude of inertia increases varies, and the current response waveform also varies. Table 8 shows the harmonic response time of the q-axis current under different inertia disturbances. When the inertia changes, the current overshoot of SMC and ISMC is the largest, the current response time of SMC is the longest, and the current waveform of ITSMC and ITFTSMC is the most stable. However, the current overshoot and response time of ITSMC are slightly larger than those of ITFTSMC. It can be seen that when the speed remains constant, the current performance of ITFTSMC remains optimal under different inertia disturbances.

**Table 8.** Comparison table of current data under inertial disturbance at static rotational speed.

| Experiment | Controller | Overshoot | Adjust Time |
|---|---|---|---|
| Q-axis current response waveform under small inertia disturbance | SMC | 35.56% | 16 ms |
| | ISMC | 30.56% | 12 ms |
| | ITSMC | 6.61% | 5 ms |
| | ITFTSMC | 5.93% | 4 ms |
| Q-axis current response waveform under moderate inertia disturbance | SMC | 46.17% | 64 ms |
| | ISMC | 45.67% | 26 ms |
| | ITSMC | 6.33% | 10 ms |
| | ITFTSMC | 6.17% | 8 ms |
| Q-axis current response waveform under large inertia disturbance | SMC | 79.54% | 145 ms |
| | ISMC | 77.58% | 33 ms |
| | ITSMC | 7.34% | 13 ms |
| | ITFTSMC | 6.97% | 12 ms |

5.3.3. Influence of Inertia Disturbance on Dynamic Rotational Speed Performance
Influence of Small Inertia Disturbance on Dynamic Speed Performance

To verify the influence of a small inertia disturbance on the dynamic rotational speed performance, the moment of inertia increases from $10^{-4}$ kg·m$^2$ to $2 \times 10^{-4}$ kg·m$^2$ at 2 s. The motor speed changed from low to medium, low to high, and medium to high. The experimental rotational speed response waveform is shown in Figure 16.

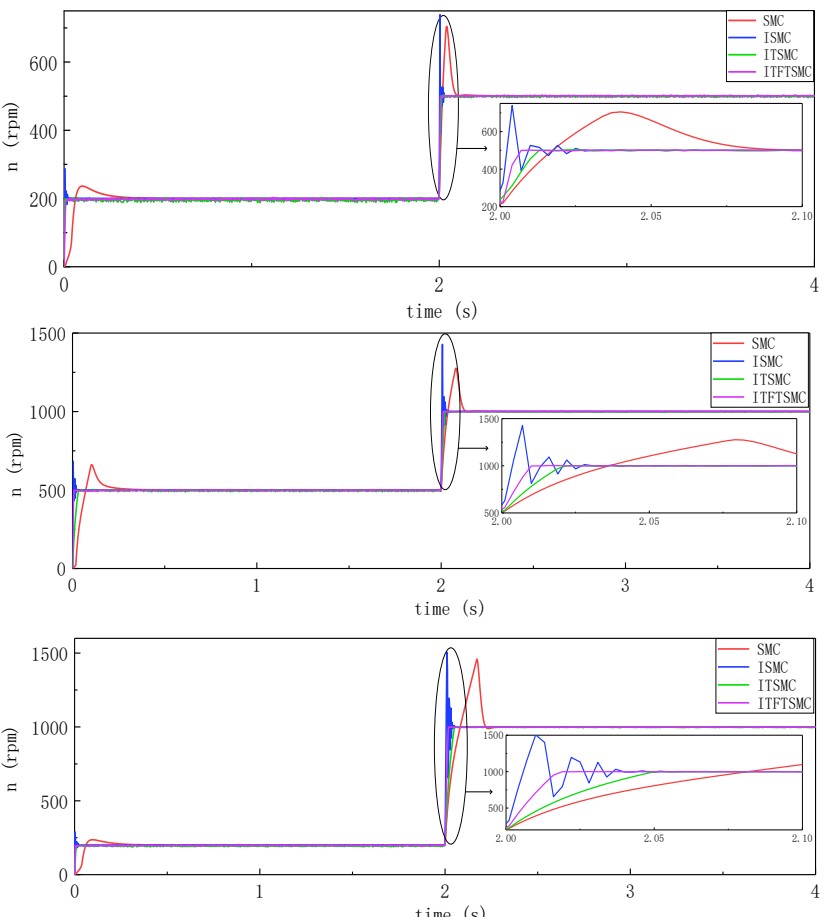

**Figure 16.** Dynamic speed response waveform under small inertia disturbances.

When the system injects small inertia interference during a change in speed, the overshoot and response time under different sliding mode controllers are shown in Table 9. The speed performance of the SMC is the worst, and its response time is more than 80 ms longer than the ITFTSMC under low, medium, and high-speed changes, and the overshoot is more than 40% larger. TSMC has a shorter response time but a higher overshoot, more than 24% higher than ITFTSMC. The response time of ITSMC is longer than that of ITFTSMC, especially when the speed changes from low speed to high speed. The response time of ITSMC is about 48 ms, and the response time of ITFTSMC is about 18 s. Clearly, when the system introduces small inertia interference during a change in speed, the ITFTSMC used herein has better anti-disturbance ability.

**Table 9.** Comparison table of dynamic speed data under small inertia changes.

| Experiment | Controller | Overshoot | Adjust Time |
|---|---|---|---|
| Dynamic speed response waveform under small inertia disturbance (low-medium speed) | SMC | 40.8% | 89 ms |
| | ISMC | 49.2% | 24 ms |
| | ITSMC | 0.39% | 12 ms |
| | ITFTSMC | 0.34% | 8 ms |
| Dynamic speed response waveform under small inertia disturbance (medium-high speed) | SMC | 47% | 124 ms |
| | ISMC | 24.3% | 29 ms |
| | ITSMC | 0.12% | 23 ms |
| | ITFTSMC | 0.11% | 12 ms |
| Dynamic speed response waveform under small inertia disturbance (low-high speed) | SMC | 50.3% | 220 ms |
| | ISMC | 48.6% | 39 ms |
| | ITSMC | 0.14% | 48 ms |
| | ITFTSMC | 0.13% | 18 ms |

Influence of Moderate Inertia Disturbance on Dynamic Rotational Speed Performance

To verify the influence of the moderate inertia volume disturbance on the dynamic rotation speed performance, the moment of inertia increases from $10^{-4}$ kg·m$^2$ to $10^{-3}$ kg·m$^2$ at 2 s. The experimental rotational speed response waveform is shown in Figure 17 below.

When the system injects moderate inertial interference when the speed changes, the overshoot and response time under different sliding mode controllers are shown in Table 10. SMC has the worst rotational speed response performance, the longest response time, and the highest overshoot. Its performance also decreases as the amplitude of rotational speed change increases. The TSMC also has a longer rotational speed response time, a higher overshoot rate, and non-stop oscillation, especially the greater the amplitude of rotational speed change, the longer the rotational speed oscillations take, and the greater the amplitude of oscillations. The overshoot of the ITSMC is similar to that of the ITFTSMC, but the response time is longer than that of the ITFTSMC. Obviously, when the system introduces moderate inertial disturbances during speed changes, the ITFTSMC used in this article has better anti-interference ability, and its rotational speed has better stability and fast performance.

Influence of Large Inertia Disturbance on Dynamic Rotational Speed Performance

To verify the influence of the large inertia volume disturbance on the dynamic rotation speed performance, the inertia volume is greatly increased, and then the moment of inertia increases from $10^{-3}$ kg·m$^2$ to $10^{-2}$ kg·m$^2$ at 2 s. The experimental rotational speed response waveform is shown in Figure 18 below.

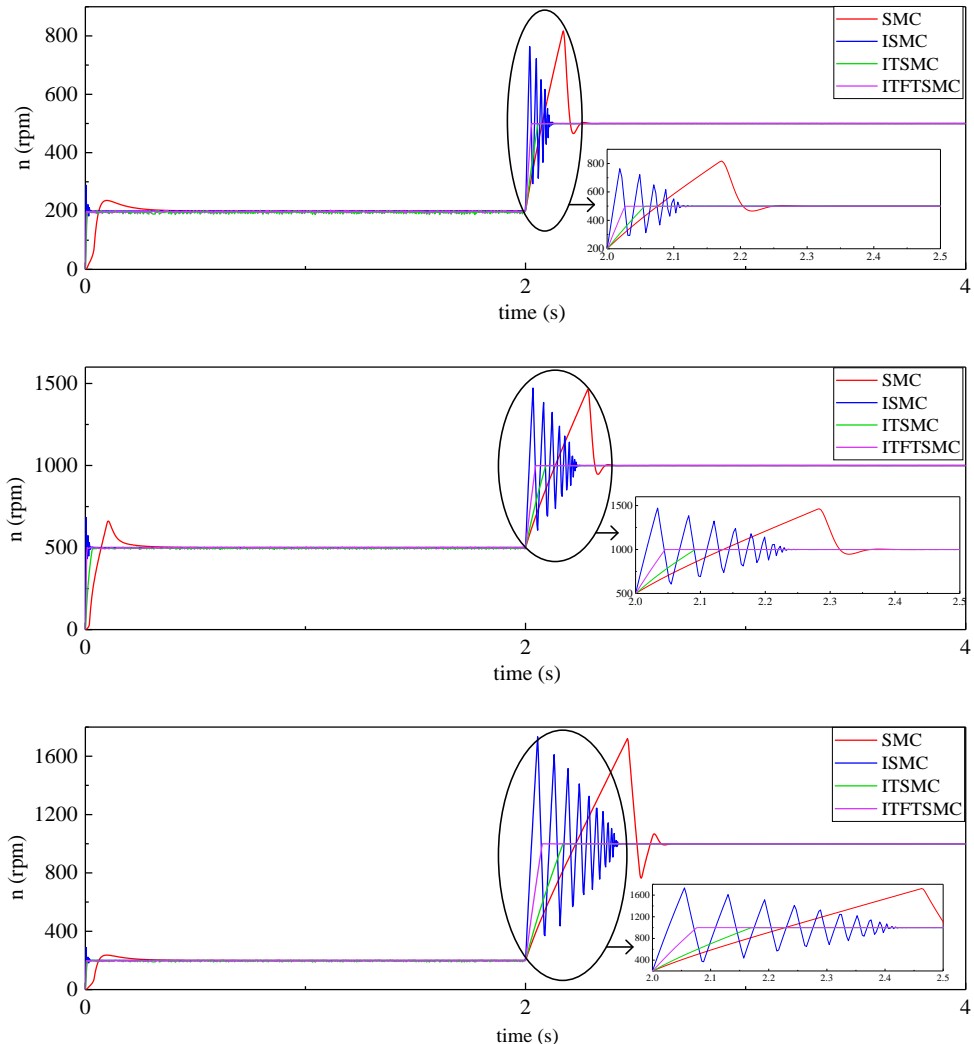

**Figure 17.** Dynamic speed response waveform under moderate inertia disturbance.

**Table 10.** Comparison table of dynamic speed data under medium inertia change.

| Experiment | Controller | Overshoot | Adjust Time |
|---|---|---|---|
| Dynamic speed response waveform under moderate inertia disturbance (low-medium speed) | SMC | 62.6% | 248 ms |
| | ISMC | 53% | 137 ms |
| | ITSMC | 0.18% | 59 ms |
| | ITFTSMC | 0.13% | 34 ms |
| Dynamic speed response waveform under moderate inertia disturbance (medium-high speed) | SMC | 48.3% | 374 ms |
| | ISMC | 48.7% | 248 ms |
| | ITSMC | 0.068% | 93 ms |
| | ITFTSMC | 0.043% | 48 ms |
| Dynamic speed response waveform under moderate inertia disturbance (low-high speed) | SMC | 64.7% | 634 ms |
| | ISMC | 66.8% | 430 ms |
| | ITSMC | 0.07% | 164 ms |
| | ITFTSMC | 0.05% | 87 ms |

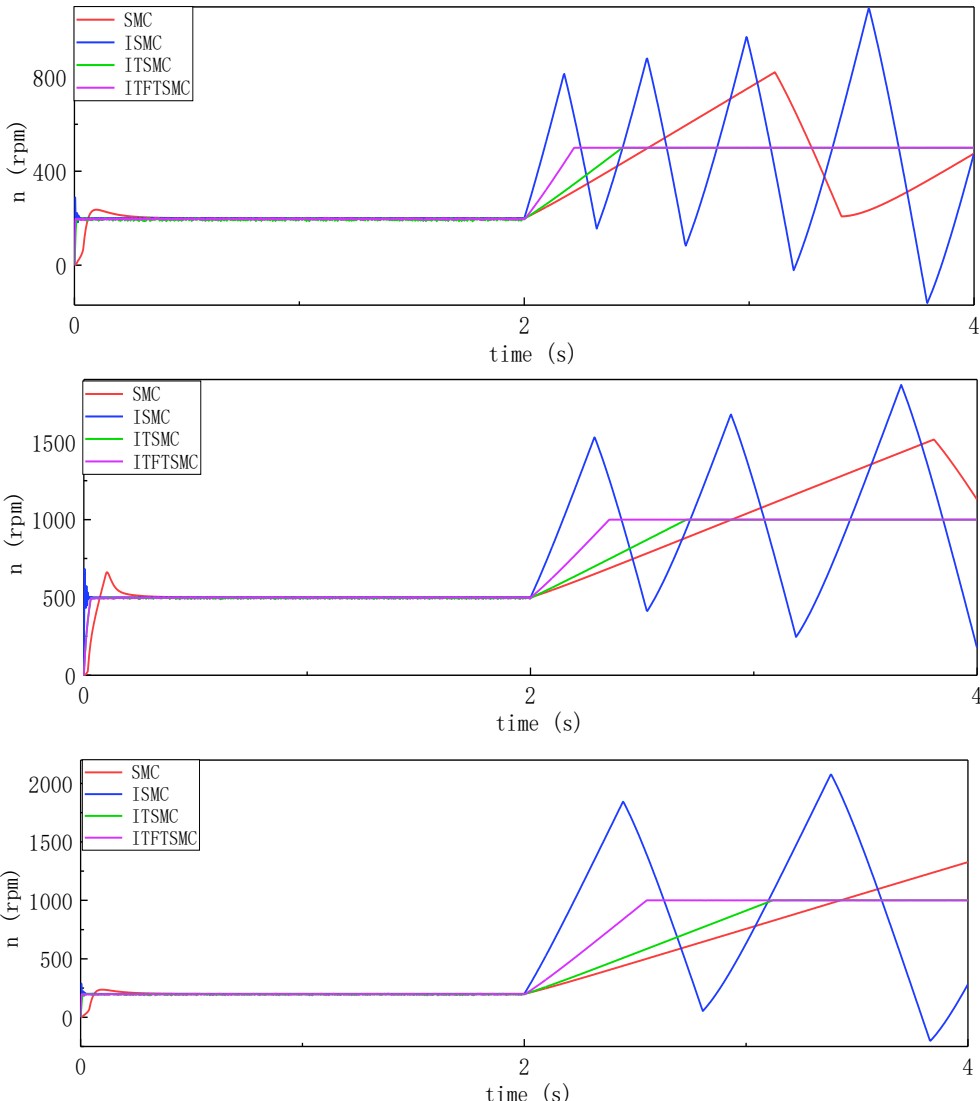

**Figure 18.** Dynamic rotational speed response waveform under large inertia disturbance.

According to Figure 18, when the system injects large inertia interference when the speed changes, the speed of SMC and TSMC is directly out of control, and the speed response time of ITSMC and ITFTSMC is also greatly increased. The greater the speed change, the greater the response time. However, no matter how the speed changes, the speed response time of ITFTSMC is smaller than the response time of ITSMC. Experiments show that even if large inertia disturbances are introduced when the motor speed changes, the ITFTSMC designed in this paper has the best anti-interference ability, and the stability and fast performance of the speed are also the best.

Effect of Inertia Disturbance on Electric Current at Dynamic Rotation Speed

To verify the effect of inertia perturbation on the current at dynamic rotation speed, this article sets the speed to increase from 200 rpm to 500 rpm and the inertia to increase from 2 s, respectively, from $10^{-4}$ kg·m$^2$ to $2 \times 10^{-4}$ kg·m$^2$, $10^{-4}$ kg·m$^2$ to $10^{-3}$ kg·m$^2$, and $10^{-3}$ kg·m$^2$ to $10^{-2}$ kg·m$^2$. Comparing the current response waveforms under these three inertia changes, Figure 19 shows the corresponding current waveforms.

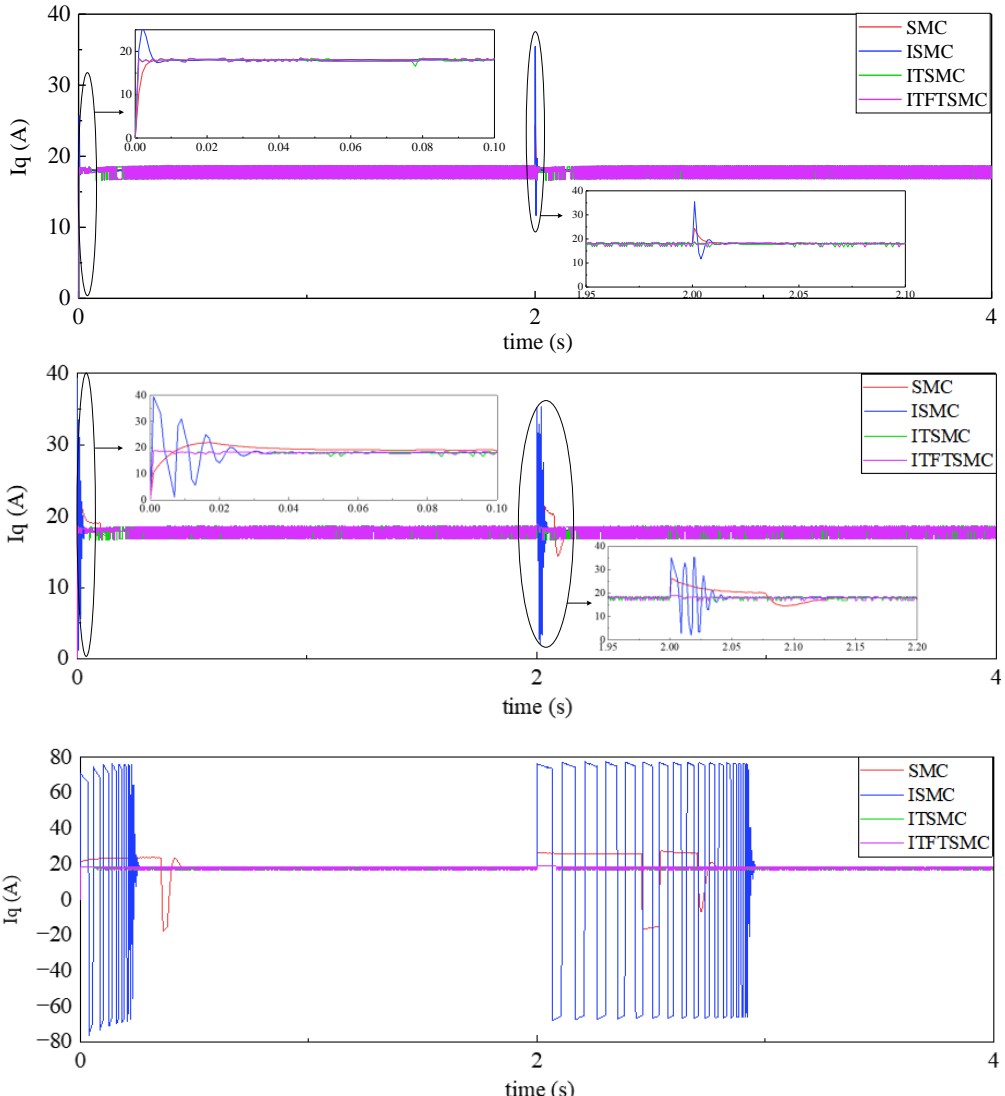

**Figure 19.** Current response waveform under inertia disturbance at dynamic rotational speed.

As shown in Figure 19, when the inertia changes, the current fluctuation at dynamic speed is worse than that at static speed. When the inertia disturbance is small, the current response time of SMC is the longest, the current overshoot of ISMC is the highest, and the current overshoot and response time of ITSMC are slightly larger than those of ITFTSMC. When the inertia disturbance is moderate, the current response time of SMC further increases, and the current fluctuation of ISMC becomes more obvious. The current overshoot and response times of ITSMC and ITFTSMC have almost not changed. When the inertia disturbance is large, the current of ISMC begins to lose control, and the current overshoot and response time of SMC are also too large. However, the current overshoot and response time of ITFTSMC remain stable. It can be seen that even if the speed changes, the current of ITFTSMC can maintain optimal performance under different inertia disturbances.

## 6. Conclusions

Based on the mathematical model of PMSM in the d-q synchronous rotating coordinate system, a method for restraining inertial disturbance is designed in this paper. By improving the extended state observer, the motor speed is observed in real time, and the observation results are fed back to the new slip-mode controller designed by the author. The sliding mode controller is added with an integral time-varying module and a fast terminal module, which can effectively improve the stability and rapidity of rotating speed,

and the improvement of the approach law also improves the fast performance of the system. The results and conclusions of this paper are as follows:

(1) Based on the mechanical motion equation of PMSM, the relationship between moment of inertia and speed change is inversely proportional, and its validity is verified by simulation;

(2) In view of the shortcomings of the recognition accuracy of the continuous model Inertia observer, this paper improves it and successfully improves the recognition accuracy;

(3) When the rotational speed of the motor encounters inertia changes while maintaining a constant value, the rotational speed will also change, and this change will increase with the increase of the inertia change amplitude. The ITFTSMC designed in this article can shorten the response time and weaken the rotational speed fluctuation when the rotational speed changes abruptly;

(4) When the rotational speed of a motor encounters an inertia change when it changes, the amplitude of the rotational speed change is greater than when it encounters an inertia change when the rotational speed remains constant. Especially when the amplitude of the rotational speed change is large, the conventional sliding mode controller has lost its function. However, the ITFTSMC designed in this paper can still stabilize the rotational speed of the motor in a relatively short time.

**Author Contributions:** Methodology, F.X. and S.N.; Software, H.W.; Formal analysis, S.N. and J.X.; Investigation, H.W. and J.X.; Resources, F.X., H.W. and Z.Z.; Writing—original draft, S.N.; Writing—review & editing, F.X. and S.N.; Visualization, Z.Z. All authors have read and agreed to the published version of the manuscript.

**Funding:** This research was funded by Natural Science Foundation of Anhui Province, grant number 2108085ME179; National Natural Science Foundation of China, grant number 51607002; Key project of National Natural Science funds, grant number 51637001.

**Data Availability Statement:** The data that support the findings of this study are available from the corresponding author upon requests.

**Conflicts of Interest:** The authors declare no conflict of interest.

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
