# Peer review of "Anti-Inertia Disturbance Control of Permanent Magnet Synchronous Motor Based on Integral Time-Varying Fast Terminal Sliding Mode"

_machines, doi:10.3390/machines11070690_

Round 1
Reviewer 1 Report (New Reviewer)
This paper proposed an integral time-varying fast terminal sliding mode control with the inertia identification, in order to improve the speed response to the inertia disturbance. The work sounds interesting and valuable, and made a certain contributions to the field. However, there are still several problems as follows:
1. The extensive editing of English language is required.
2. The introduction provides sufficient background about different methods, but provides only a little about the sliding mode control. It is suggested that the introduction should focus on the sliding mode control itself.
3. The improvement or creativeness of the proposed ITFTSMC method is not demonstrated clearly in the paper. Is this method a novel one? Or is this method improved according to the relevant research?
4. No current waveforms in the validation. Although this paper cares more about the speed response, the current response is more critical to the system characteristics. The current response can reveal the torque response more clearly, and the torque directly determines the speed response. It is suggested that current waveforms such as id and iq be supplemented in the validation.
5. No sufficient data analysis in the validation. It is suggested that more data analysis such as overshoot and adjust time, especially the data table for comparison, be supplemented in the validation.
6. No values of key parameters provided, e.g. the parameters d, l, b1, b2, etc. in the ESO, the parameters c, a, b, r, p, q, etc. in the SMC. It is suggested that the key parameters be supplemented prior to the validation.
7. How does the system work when the torque changes?
1. The extensive editing of English language is required.
Author Response
Response to Reviewers’ Comments for Reviewer #1
Anti-inertia Disturbance Speed Control of Permanent Magnet Synchronous Motor Based on Integral Time-varying Fast Terminal Sliding Mode
The authors would like to thank the reviewers for very constructive and detailed comments that have helped us improve the paper. All the comments and suggestions have been carefully considered. All changes to the manuscript have been highlighted with the BLUE color. The following summarizes our revisions and explanations itemized according to each reviewer’s comments. Please note unless otherwise specified, all the page numbers cited below are for the revised manuscript. Especially, please check the text in BLUE color in the mentioned pages in the following responses.
Reviewer’s Comment:
- The extensive editing of English language is required.
Authors’ Response:
Thanks for the reviewer’s valuable comments.The authors have made the modifications accordingly.
- The introduction provides sufficient background about different methods, but provides only a little about the sliding mode control. It is suggested that the introduction should focus on the sliding mode control itself.
Authors’ Response:
The authors are grateful to the reviewer for the constructive comments that have helped us improve the paper.Here is the derivation process I have modified:
The note [21] proposes a robust integral sliding mode (RISM) manifold and a corresponding design method for stabilization control for uncertain systems with control input time delay. An improved sliding mode control, which is delay-dependent and suitable for small input time delay, keeps system stay on the neighborhood of the RISM surface in finite time and improving the system stability.Paper [22] studies fault-tolerant control (FTC) designs based on nonsingular terminal sliding-mode control and nonsingular fast terminal sliding-mode control (NFTSMC). The proposed active FTC laws are shown to be able to achieve fault-tolerant objectives and maintain stabilization performance even when some of the actuators fail to operate.In order to achieve a robust tracking performance, a novel sliding-mode control law [23] is designed in the SMC unit, and the adaptive unit is put forward to deal with uncertain gain.The article [24] proposes a continuous nonsingular terminal sliding-mode control with integral-type sliding surface (CNTSMC-ISS) framework for disturbed systems. Compared with the existing sliding-mode controllers, the noteworthy contributions are the alleviation of the chattering phenomenon, the fast finite-time stability, and the singularity-free and ease-of-implementation characteristics.(Page 2,line 28--line 42)
- The improvement or creativeness of the proposed ITFTSMC method is not demonstrated clearly in the paper. Is this method a novel one? Or is this method improved according to the relevant research?
Authors’ Response:
Thanks for the reviewer’s valuable comments.The author has read a large number of books on sliding mode control and learned about the advantages and disadvantages of integral sliding mode control, time-varying sliding mode control, and fast terminal sliding mode control. In order to compensate for the shortcomings of a single sliding mode controller, these three sliding mode controllers are combined into a new type of sliding mode controller to complement each other. On this basis, in order to further improve the stability of the sliding mode controller, the author also made improvements to the reaching law.
Third, to solve the motor speed fluctuation caused by the sudden inertia change, an integral time-varying fast terminal sliding mode control method is proposed, which improves both speed stability through integral time-varying and rapidity through the fast terminal. (Page 1,line 11--line 14)
A new convergence law was defined as shown in Equation :(Page 6,line 9)
- No current waveforms in the validation. Although this paper cares more about the speed response, the current response is more critical to the system characteristics. The current response can reveal the torque response more clearly, and the torque directly determines the speed response. It is suggested that current waveforms such as id and iq be supplemented in the validation.
Authors’ Response:
Thanks for the reviewer’s valuable comments.Please allow me to express my sincere apologies to you. The author only provided a set of current waveform diagrams. After multiple experiments, the author found that the overall trend of the current waveform and the speed waveform is the same. Adding 18 sets of current waveforms that are similar to the speed waveform is not significant and can also lead to a bulky paper. All authors only retained one set of current waveform, as shown below.
Figure 4 shows the comparison waveform of the q-axis current caused by conventional sliding mode control and integral sliding mode control.
Figure 4. Iq contrast waveform of SMC and ISMC.
As shown in Figure 3, the speed of SMC has a large overshoot and long response time when the motor starts and the inertia changes, and the speed waveform obtained by the sliding mode controller after adding the integral module has a great improvement in response time. However, the overshoot remains large and needs further improvement.As shown in Figure 4, the fluctuation trend of the q-axis current and the fluctuation trend of the speed are roughly the same, but the current fluctuation amplitude of ISMC is extremely large, which is not conducive to the operation of the motor and needs further improvement.
(Page 7,line 1--line 11)
Figure 6 shows the q-axis current comparison waveform under SMC, ISMC, and ITSMC.
Figure 6. Iq contrast waveform of SMC , ISMC and ITSMC.
Following Figure 5, the speed of ISMC has a large overshoot when the motor is started, and the speed waveform obtained by the sliding mode controller after adding the time-varying module has been greatly improved in the overshoot.As shown in Figure 6, the overall current fluctuation of ITSMC slightly increases, but the overshoot is much smaller than that of ISMC, and there is still room for improvement.(Page 7,line 17--line 14)
Figure 8 shows the q-axis current comparison waveform under SMC, ISMC, ITSMC and ITFTSMC.
Figure 8. Iq contrast waveform of SMC , ISMC,ITSMC and ITFTSMC.
As shown in Figure 7, after the introduction of the fast terminal module in ITSMC, the response time of the motor during startup is significantly reduced, and there is almost no overshoot. Compared to the other three sliding mode controllers, ITFTSMC provides a faster and more stable response speed during motor startup. When the moment of inertia changes, both SMC and ISMC have larger overshoots and longer response times, while both ITSMC and ITFTSMC have smaller response times and almost no overshoots. Overall,When the moment of inertia suddenly changes, the rotational speed of the ITFTSMC has the best fast and stable performance.And as shown in Figure 8, the q-axis current of ITFTSMC also has the same characteristics, with the best response time and overshoot, which meets the design requirements.(Page 8,line 6--line 19)
5.No sufficient data analysis in the validation. It is suggested that more data analysis such as overshoot and adjust time, especially the data table for comparison, be supplemented in the validation.
Authors’ Response:
Thanks for the reviewer’s valuable comments. The comparison data table for overshoot and response time has been added, and the modifications are as follows:
For easier comparison and illustration, the overshoot and response time of different sliding mode controllers during motor startup under low, medium, and high speed conditions are provided to evaluate the effectiveness of the ITFTSMC method. The data comparison is shown in Table 2.Among them, SMC fluctuates the most: the speed overshoot at low, medium, and high speeds is more than 15% higher than ITFTSMC, and the response time is more than 0.3s longer than ITFTSMC. The response time of TSMC is very short, although the overshoot is more than 30% higher than that of ITFTSMC. ITSMC generates more fluctuations at low speeds than ITFTSMC, and its response time at high speeds is about 75ms longer than ITFTSMC. Obviously, the speed stability and fast performance of ITFTSMC during motor startup are superior to the other three controllers.
Table 2.Data comparison table during motor startup
|
experiment |
controller |
overshoot |
adjust time |
|
Speed response waveform when the motor starts(low speed) |
SMC |
16.5% |
407ms |
|
ISMC |
42% |
43ms |
|
|
ITSMC |
3.5% |
18ms |
|
|
ITFTSMC |
2% |
16ms |
|
|
Speed response waveform when the motor starts (medium speed) |
SMC |
32% |
326ms |
|
ISMC |
37% |
48ms |
|
|
ITSMC |
4.7% |
13ms |
|
|
ITFTSMC |
4.2% |
12ms |
|
|
Speed response waveform when the motor starts(high speed) |
SMC |
59% |
402ms |
|
ISMC |
57% |
63ms |
|
|
ITSMC |
5.6% |
113ms |
|
|
ITFTSMC |
4.8% |
38ms |
(Page 11,line 2--line 12)
When the rotational inertia increases from to, the overshoot and response time under different sliding mode controllers are shown in Table 3.The rotational speed of a conventional SMC changes most significantly,with the highest overshoot and the longest response time; ITFTSMC has the smallest speed fluctuation and the shortest response time. Under low, medium, and high speed conditions, the speed fluctuation of conventional SMC is about 20 rpm higher than that of ITFTSMC, and the response time is about 0.1 s slower; The speed fluctuation of TSMC is about 4 rpm higher than that of ITFTSMC, and the response time is about 20ms slower; The rotational speed response time of ITSMC is similar to that of ITFTSMC, but the overall fluctuation of its rotational speed is greater than that of ITFTSMC. Experiments have proven that when the inertia changes in a small range, the rotational speed of ITFTSMC has the best stability and fast performance at both low and medium and high speeds.
Table 3.Comparison Table of Static Speed Data under Small Inertia Change
|
experiment |
controller |
overshoot |
adjust time |
|
Static rotational speed response waveform under small inertia disturbance(low speed) |
SMC |
11% |
107ms |
|
ISMC |
2% |
26ms |
|
|
ITSMC |
0.34% |
8ms |
|
|
ITFTSMC |
0.27% |
7ms |
|
|
Static rotational speed response waveform under small inertia disturbance(medium speed) |
SMC |
4% |
112ms |
|
ISMC |
0.8% |
27ms |
|
|
ITSMC |
0.073% |
7.8ms |
|
|
ITFTSMC |
0.058% |
6.9ms |
|
|
Static rotational speed response waveform under small inertia disturbance(high speed) |
SMC |
2% |
124ms |
|
ISMC |
0.47% |
27ms |
|
|
ITSMC |
0.038% |
8ms |
|
|
ITFTSMC |
0.029% |
7ms |
(Page 12,line 9--line 12;Page 13,line 1--line 8)
When the moment of inertia increases fromto ,the overshoot and response time under different sliding mode controllers are shown in Table 4. The SMC speed change remains the largest and the ITFTSMC speed change remains the smallest. Under low, medium, and high speed conditions, SMC speed overshoot is more than 15% higher than ITFTSMC,and the response time is more than 110ms longer than ITFTSMC; TSMC overshoot is more than 1.5% higher than ITFTSMC, and the response time is more than 25ms longer than ITFTSMC; The rotational speed fluctuation caused by ITSMC before and after the inertia mutation is greater than that of ITFTSMC, and the rotational speed fluctuation at high speed is the most obvious. Experiments have shown that when inertia changes in a medium range, its rotational speed still has the best stability and fast performance.
Table 4.Comparison Table of Static Speed Data under Medium Inertia Change
|
experiment |
controller |
overshoot |
adjust time |
|
Static rotational speed response waveform under moderate inertia disturbance(low speed) |
SMC |
87.5% |
125ms |
|
ISMC |
12.5% |
27ms |
|
|
ITSMC |
4% |
1.2ms |
|
|
ITFTSMC |
2.5% |
1.1ms |
|
|
Static rotational speed response waveform under moderate inertia disturbance(medium speed) |
SMC |
35% |
120ms |
|
ISMC |
5% |
26ms |
|
|
ITSMC |
1.6% |
3.4ms |
|
|
ITFTSMC |
1.5% |
2.9ms |
|
|
Static rotational speed response waveform under moderate inertia disturbance(high speed) |
SMC |
17.5% |
118ms |
|
ISMC |
2.5% |
27ms |
|
|
ITSMC |
1.3% |
1.4ms |
|
|
ITFTSMC |
0.89% |
0.83ms |
(Page 14,line 2--line 12)
When the moment of inertia increases fromto , the overshoot and response time under different sliding mode controllers are shown in Table 5.The rotational speed of the SMC changes most acutely. Under low, medium, and high speed conditions, its rotational speed increases by about 90 rpm, and the response time also exceeds 0.5s; The rotation speed fluctuation of ISMC is relatively small, but the rotation speed increases by about 20 rpm; The response time also reached about 0.1s; The rotational speed fluctuations of ITSMC and ITFTSMC are minimal, but considering that the rotational speed fluctuations of ITSMC before the inertia change are greater than the rotational speed fluctuations of ITFTSMC, it can be concluded that when the inertia changes over a wide range, the rotational speed of ITFTSMC still has the best stability and fast performance.
Table 5.Comparison Table of Static Speed Data under Large Inertia Change
|
experiment |
controller |
overshoot |
adjust time |
|
Static rotational speed response waveform under large inertia disturbance(low speed) |
SMC |
45% |
560ms |
|
ISMC |
9.8% |
123ms |
|
|
ITSMC |
6% |
4.2ms |
|
|
ITFTSMC |
4.3% |
3.7ms |
|
|
Static rotational speed response waveform under large inertia disturbance(medium speed) |
SMC |
18% |
556ms |
|
ISMC |
4% |
118ms |
|
|
ITSMC |
2.4% |
3.8ms |
|
|
ITFTSMC |
1.4% |
3.9ms |
|
|
Static rotational speed response waveform under large inertia disturbance(high speed) |
SMC |
9% |
547ms |
|
ISMC |
2% |
107ms |
|
|
ITSMC |
1.3% |
4ms |
|
|
ITFTSMC |
0.83% |
3.9ms |
(Page 15,line 2--line 11)
When the system injects small inertia interference during a change in speed,the overshoot and response time under different sliding mode controllers are shown in Table 6. The speed performance of the SMC is the worst, and its response time is more than 80ms longer than ITFTSMC under low, medium, and high-speed changes, and the overshoot is more than 40% larger. TSMC has a shorter response time, but a higher overshoot, more than 24% higher than ITFTSMC. The response time of ITSMC is longer than that of ITFTSMC, especially the speed changes from low speed to high speed, the response time of ITSMC is about 48ms, and the response time of ITFTSMC is about 18s. Clearly, when the system introduces small inertia interference during a change in speed, the ITFTSMC used herein has better anti-disturbance ability.
Table 6.Comparison Table of Dynamic Speed Data under Small Inertia Change
|
experiment |
controller |
overshoot |
adjust time |
|
Dynamic speed response waveform under small inertia disturbance(low-medium speed) |
SMC |
40.8% |
89ms |
|
ISMC |
49.2% |
24ms |
|
|
ITSMC |
0.39% |
12ms |
|
|
ITFTSMC |
0.34% |
8ms |
|
|
Dynamic speed response waveform under small inertia disturbance(medium-high speed) |
SMC |
47% |
124ms |
|
ISMC |
24.3% |
29ms |
|
|
ITSMC |
0.12% |
23ms |
|
|
ITFTSMC |
0.11% |
12ms |
|
|
Dynamic speed response waveform under small inertia disturbance(low-high speed) |
SMC |
50.3% |
220ms |
|
ISMC |
48.6% |
39ms |
|
|
ITSMC |
0.14% |
48ms |
|
|
ITFTSMC |
0.13% |
18ms |
(Page 16,line 6--line 16)
When the system injects moderate inertial interference when the speed changes,the overshoot and response time under different sliding mode controllers are shown in Table 7. SMC has the worst rotational speed response performance, the longest response time, and the highest overshoot. Its performance also decreases as the amplitude of rotational speed change increases; The TSMC also has a longer rotational speed response time, a higher overshoot rate, and non-stop oscillation, especially the greater the amplitude of rotational speed change, the longer the rotational speed oscillations take, and the greater the amplitude of oscillations; The overshoot of ITSMC is similar to that of ITFTSMC, but the response time is longer than that of ITFTSMC. Obviously, when the system introduces moderate inertial disturbances during speed changes, the ITFTSMC used in this article has better anti-interference ability, and its rotational speed has better stability and fast performance.
Table 7.Comparison Table of Dynamic Speed Data under Medium Inertia Change
|
experiment |
controller |
overshoot |
adjust time |
|
Dynamic speed response waveform under moderate inertia disturbance(low-medium speed) |
SMC |
62.6% |
248ms |
|
ISMC |
53% |
137ms |
|
|
ITSMC |
0.18% |
59ms |
|
|
ITFTSMC |
0.13% |
34ms |
|
|
Dynamic speed response waveform under moderate inertia disturbance(medium-high speed) |
SMC |
48.3% |
374ms |
|
ISMC |
48.7% |
248ms |
|
|
ITSMC |
0.068% |
93ms |
|
|
ITFTSMC |
0.043% |
48ms |
|
|
Dynamic speed response waveform under moderate inertia disturbance(low-high speed) |
SMC |
64.7% |
634ms |
|
ISMC |
66.8% |
430ms |
|
|
ITSMC |
0.07% |
164ms |
|
|
ITFTSMC |
0.05% |
87ms |
(Page 17,line 6--line 16)
- No values of key parameters provided, e.g. the parameters d, l, b1, b2, etc. in the ESO, the parameters c, a, b, r, p, q, etc. in the SMC. It is suggested that the key parameters be supplemented prior to the validation.
Authors’ Response:
Thanks for the reviewer’s valuable comments.The author provides the specific parameter values as follows:
where,, and ;(Page 3,line 11)
where, ;(Page 3,line 25)
Where,,, .(Page 6,line 2)
Where,.(Page 6,line 11)
- How does the system work when the torque changes?
Authors’ Response:
Thanks for the reviewer’s valuable comments.As shown in the figure below, a change in inertia will drive a change in current on the one hand, and on the other hand, it will drive a change in motor speed, resulting in motor speed error. Through the current change and speed error, the Moment of inertia observation value can be obtained by using the extended State observer, and the reference current input to the Current loop can be obtained by substituting the speed error and inertia observation value into the ITFTSMC, thus forming a complete closed-loop control.
Reviewer 2 Report (New Reviewer)
Anti inertia Disturbance Control of Permanent Magnet Synchronous Motor Based on Integral Time-varying Fast Terminal Sliding Mode
Review
This paper presents a method for the speed control of a permanent magnet synchronous motor, during disturbances of the moment of inertia. To solve the motor speed fluctuation caused by the disturbances of inertia, an integral time-varying fast terminal sliding mode control method is proposed, which improves the speed stability. Simulation and experiments were used to verify the feasibility and effectiveness of the proposed method.
Is this reviewer opinion that the paper is interesting and describe a useful speed control method. The manuscript contains modeling, simulation, and a laboratory test bench with results.
However, some changes are proposed.
The authors must use the Microsoft Word template to prepare the manuscript and follow the rules of the journal. The equations must be written using a mathematical software.
All equations, because most are retrieved from literature, need exact references.
The software for solving the model and obtaining all figures must be explained.
Regarding the experimental platform, there are necessary and must be explained the details of all components, modules, abbreviations, codes. The nominal power and nominal inertia of the motor must be given.
The authors must add a section describing a real application, with industrial motors, with real data from process, where could be used the proposed model and the solution.
Conclusions must be focused on applying the proposed method to real industrial plants.
The language must be edited for syntax and technical terminology.
Please see above
Author Response
Response to Reviewers’ Comments for Reviewer #2
Anti-inertia Disturbance Control of Permanent Magnet Synchronous Motor Based on Integral Time-varying Fast Terminal Sliding Mode
The authors would like to thank the reviewers for very constructive and detailed comments that have helped us improve the paper. All the comments and suggestions have been carefully considered. All changes to the manuscript have been highlighted with the BLUE color. The following summarizes our revisions and explanations itemized according to each reviewer’s comments. Please note unless otherwise specified, all the page numbers cited below are for the revised manuscript. Especially, please check the text in BLUE color in the mentioned pages in the following responses.
Reviewer’s Comment:
This paper presents a method for the speed control of a permanent magnet synchronous motor, during disturbances of the moment of inertia. To solve the motor speed fluctuation caused by the disturbances of inertia, an integral time-varying fast terminal sliding mode control method is proposed, which improves the speed stability. Simulation and experiments were used to verify the feasibility and effectiveness of the proposed method.
Is this reviewer opinion that the paper is interesting and describe a useful speed control method. The manuscript contains modeling, simulation, and a laboratory test bench with results.
However, some changes are proposed.
The authors must use the Microsoft Word template to prepare the manuscript and follow the rules of the journal. The equations must be written using a mathematical software.
All equations, because most are retrieved from literature, need exact references.
The software for solving the model and obtaining all figures must be explained.
Regarding the experimental platform, there are necessary and must be explained the details of all components, modules, abbreviations, codes. The nominal power and nominal inertia of the motor must be given.
The authors must add a section describing a real application, with industrial motors, with real data from process, where could be used the proposed model and the solution.
Conclusions must be focused on applying the proposed method to real industrial plants.
The language must be edited for syntax and technical terminology.
Authors’ Response:
The authors thank the reviewer for the comment.
The author used a Microsoft Word template to prepare the manuscript and followed the journal's rules. The equations are also written using mathematical software and ensure their authenticity.
The solving model and simulation testing of this article are based on Simulink simulation software, which is a module diagram environment used for multi domain simulation and model based design. It supports system design, simulation, automatic code generation, and continuous testing and validation of embedded systems. Simulink can be modeled with continuous sampling time, discrete sampling time, or two mixed sampling times, and it also supports multi rate systems, where different parts of the system have different sampling rates.
The author has improved the explanation of the relevant details of the experimental platform and provided the nominal power and inertia of the motor:
The platform is mainly comprised of 8 parts: a permanent magnet synchronous motor; a magnetic powder clutch; a torque sensor; a half-shaft bushing; CSPACE;2500PPR incremental encoder; oscilloscope; and motor drive circuit board. Among these, the change of PMSM moment of inertia is realized by the increase and decrease of the half-shaft casing. Encoders are then used for the collection of motor speeds, and CSPACE is a semi-physical simulation system based on TMS320F28335DSP.It has AD, DA, IO, Encoder,PWM, and other simulation functions.After the control algorithm has been designed in MATLAB/Simulink, the DSP code can be generated and the corresponding control signals can be generated.(Page 9,line 11--line 18)
Table 1. PMSM parameters.
I would like to apologize to you, the method proposed by the author has not yet been applied to actual industrial factories, only to existing motors in the laboratory. The author is purchasing a robotic arm and applying the proposed method to it, but has not yet received the goods and cannot complete your suggestion within the deadline. I am truly sorry.
Round 2
Reviewer 1 Report (New Reviewer)
Response 3: Where or how have the proposed ITFTSMC been improved? It is still suggested the improvement or creativeness of the proposed ITFTSMC should be emphasized on or clearly pointed out in the end of the introduction, 4.1. ITFTSMC analytical model and the conclusion, especially in the 4.1. ITFTSMC analytical model.
Response 4: It is still suggested that appropriate current or torque waveforms be supplemented in 5. Experimental validation.
Response 6: Now, the parameters are listed separately in different sections. It is still suggested that the key parameters be supplemented as a table in 5. Experimental validation.
Author Response
Response to Reviewers’ Comments for Reviewer #1
Anti-inertia Disturbance Speed Control of Permanent Magnet Synchronous Motor Based on Integral Time-varying Fast Terminal Sliding Mode
The authors would like to thank the reviewers for very constructive and detailed comments that have helped us improve the paper. All the comments and suggestions have been carefully considered. All changes to the manuscript have been highlighted with the BLUE color. The following summarizes our revisions and explanations itemized according to each reviewer’s comments. Please note unless otherwise specified, all the page numbers cited below are for the revised manuscript. Especially, please check the text in BLUE color in the mentioned pages in the following responses.
Reviewer’s Comment:
- Response 3: Where or how have the proposed ITFTSMC been improved? It is still suggested the improvement or creativeness of the proposed ITFTSMC should be emphasized on or clearly pointed out in the end of the introduction, 4.1. ITFTSMC analytical model and the conclusion, especially in the 4.1. ITFTSMC analytical model..
Authors’ Response:
Thanks for the reviewer’s valuable comments.The authors have made the modifications accordingly.Here is the derivation process I have modified:
In order to improve the tracking accuracy of the system to the motor speed, this paper improves the extended state observer to a new inertia observer. By adding a time-varying function, the gain increases slowly, reducing the impact on the motor speed while observing the inertia. To improve the stability and rapidity of the motor speed when the inertia changes, this paper proposes an integral time-varying fast terminal sliding mode control (ITFTSMC) method. In order to shorten the duration of the approaching motion stage and ensure the dynamic quality of the approaching motion, the author improves the exponential approaching law and proposes a new type of approaching law; In order to improve the stability and fast performance of the motor speed, the author designed a new type of integral time-varying module and fast terminal module to improve the conventional sliding mode controller. The integral time-varying module can improve the stability of the motor speed, keep the speed overshoot within a small error range, and the fast terminal module can improve the speed performance of the motor speed, so that the speed can recover to stability in a short time. Finally, experiments were designed to verify the effectiveness of the method.(Page 2,line 45--line 46,Page 3,line 1--line 12)
In order to improve the fast and stable performance of the rotating speed of the motor, a new sliding mode surface is defined in this paper by combining the integral time-varying module and the fast terminal module. The function of the sliding mode surface is shown in Equation(13)
(13)
(Page 5,line 21--line 22,Page 6,line 1--line 3)
In order to shorten the time of approaching motion and improve the dynamic quality of approaching motion, this paper improves the exponential approach law and designs a new approach law, as shown in Equation(16):
(16)
(Page 6,line 11--line 14)
Based on the mathematical model of PMSM in d-q synchronous rotating coordinate system, a method for restraining inertial disturbance is designed in this paper. By improving the extended state observer, the motor speed is observed in real time, and the observation results are fed back to the new slip-mode controller designed by the author. The sliding mode controller is added with an integral time-varying module and a fast terminal module, which can effectively improve the stability and rapidity of rotating speed, and the improvement of the approach law also improves the fast performance of the system. The results and conclusions of this paper are as follows.
(Page 22,line 6--line 13)
2.Response 4: It is still suggested that appropriate current or torque waveforms be supplemented in 5. Experimental validation.
Authors’ Response:
The authors are grateful to the reviewer for the constructive comments that have helped us improve the paper.Here is the derivation process I have modified:
5.3.2.4.Effect of inertia disturbance on electric current at static rotation speed
To verify the effect of inertia perturbation on the current at static speed, this article sets the speed to 500rpm, and the inertia increases from 2s, respectively fromto,to, andto. Comparing the current response waveforms under these three inertia changes, Figure 15 shows the corresponding current waveforms.
Figure 15. Current response waveform under inertia disturbance at static rotational speed
The amplitude of inertia increase varies, and the current response waveform also varies. Table 8 shows the harmonic response time of the q-axis current under different inertia disturbances. When the inertia changes, the current overshoot of SMC and ISMC is the largest, the current response time of SMC is the longest, and the current waveform of ITSMC and ITFTSMC is the most stable. However, the current overshoot and response time of ITSMC are slightly larger than those of ITFTSMC. It can be seen that when the speed remains constant, the current performance of ITFTSMC remains optimal under different inertia disturbances.
Table 8.Comparison Table of Current Data under Inertial Disturbance at Static Rotational Speed
|
experiment |
controller |
overshoot |
adjust time |
|
Q-axis current response waveform under small inertia disturbance |
SMC |
35.56% |
16ms |
|
ISMC |
30.56% |
12ms |
|
|
ITSMC |
6.61% |
5ms |
|
|
ITFTSMC |
5.93% |
4ms |
|
|
Q-axis current response waveform under moderate inertia disturbance |
SMC |
46.17% |
64ms |
|
ISMC |
45.67% |
26ms |
|
|
ITSMC |
6.33% |
10ms |
|
|
ITFTSMC |
6.17% |
8ms |
|
|
Q-axis current response waveform under large inertia disturbance |
SMC |
79.54% |
145ms |
|
ISMC |
77.58% |
33ms |
|
|
ITSMC |
7.34% |
13ms |
|
|
ITFTSMC |
6.97% |
12ms |
(Page 16,line 10--line 15,Page 17,line 1--line 9)
5.3.3.4.Effect of inertia disturbance on electric current at dynamic rotation speed
To verify the effect of inertia perturbation on the current at dynamic rotation speed, this article sets the speed to increase from 200rpm to 500rpm, and the inertia increases from 2s, respectively fromto,to, andto. Comparing the current response waveforms under these three inertia changes, Figure 19 shows the corresponding current waveforms.
Figure 19. Current response waveform under inertia disturbance at dynamic rotational speed
As shown in Figure 19, when the inertia changes, the current fluctuation at dynamic speed is worse than that at static speed. When the inertia disturbance is small, the current response time of SMC is the longest, the current overshoot of ISMC is the highest, and the current overshoot and response time of ITSMC are slightly larger than those of ITFTSMC.When the inertia disturbance is moderate, the current response time of SMC further increases, and the current fluctuation of ISMC becomes more obvious. The current overshoot and response time of ITSMC and ITFTSMC have almost no change.When the inertia disturbance is large, the current of ISMC begins to lose control, and the current overshoot and response time of SMC are also too large. However, the current overshoot and response time of ITFTSMC remain stable. It can be seen that even if the speed changes, the current of ITFTSMC can maintain optimal performance under different inertia disturbances.
(Page 21,line 9--line 22,Page 22,line 1--line 4)
- Response 6: Now, the parameters are listed separately in different sections. It is still suggested that the key parameters be supplemented as a table in 5. Experimental validation.
Authors’ Response:
The authors are grateful to the reviewer for the constructive comments that have helped us improve the paper.Here is the derivation process I have modified:
Table 2. shows the key parameters of the inertia observer.
Table 2. parameters of the inertia observer.
Table 3. shows the key parameters of ITFTSMC.
Table 3. parameters of ITFTSMC.
(Page 10,line 7--line 10)

Reviewer 2 Report (New Reviewer)
Following up with previous suggestions, the manuscript can advance for publication.
Minor editing before publication .
Author Response
Response to Reviewers’ Comments for Reviewer #2
Anti-inertia Disturbance Speed Control of Permanent Magnet Synchronous Motor Based on Integral Time-varying Fast Terminal Sliding Mode
The authors would like to thank the reviewers for very constructive and detailed comments that have helped us improve the paper. All the comments and suggestions have been carefully considered.
Reviewer’s Comment:
- Following up with previous suggestions, the manuscript can advance for publication.
Authors’ Response:
The author is very pleased to receive recognition from the reviewers. Please allow the author to express sincere gratitude to the reviewers. The authors have completed the revisions again and will definitely live up to your expectations.
This manuscript is a resubmission of an earlier submission. The following is a list of the peer review reports and author responses from that submission.
Round 1
Reviewer 1 Report
Please, in the resuts chapter, mention the sampling time of control (minimum-maximum if not constant) and the switching frequency of the inverter (minimum-maximum if the SVPWM is not of the constant switching frequency type).
Author Response
Response to Reviewers’ Comments for Reviewer #1
ISSN 2075-1702: Anti-inertia Disturbance Speed Control of Permanent Magnet Synchronous Motor Based on Integral Time-varying Fast Terminal Sliding Mode
The authors would like to thank the reviewers for very constructive and detailed comments that have helped us improve the paper. All the comments and suggestions have been carefully considered. All changes to the manuscript have been highlighted with the BLUE color. The following summarizes our revisions and explanations itemized according to each reviewer’s comments. Please note unless otherwise specified, all the page numbers cited below are for the revised manuscript. Especially, please check the text in BLUE color in the mentioned pages in the following responses.
Reviewer’s Comment:
1.Please, in the resuts chapter, mention the sampling time of control (minimum-maximum if not constant) and the switching frequency of the inverter (minimum-maximum if the SVPWM is not of the constant switching frequency type).
Authors’ Response:
The authors thank the reviewer for the comment.The control sampling time is 0.01s, and the switching frequency of the inverter is 5kHz.

Reviewer 2 Report
In control theory, the moment of inertia is a parameter and not a disturbance signal. Therefore, the very clunky title is completely unclear. The literature on robust or insensitive controller design is very broad and should be used as a reference.
A notation of the equations such as (2) is completely unreadable. The difference in the length of the fractional dash is difficult to see. Please use standard methods of mathematical typesetting.
Chapter 2 discusses the well-known simplified equation of motion dynamics. It contributes nothing of interest. Why did the mean value of the speed change after the change in inertia ?
The Landau observer is not widely known and should be briefly introduced. Unfortunately, the structure of Chapter 3 is very unclear. How are the coefficients leading to equation (13) of the nature of the PI member selected on the basis of the equations quoted. It is not clearly stated which tests are performed (set speed step?) and which signals are measured for inertia identification.
What explains the assumption of constant velocity set in equation (16)?
Figures 5, 6, 7 are completely unreadable. The time scale is selected incorrectly, nothing happens 80% of the time in the figure
The stability analysis in no way addresses the range of variability of J and the other parameters.
Why in the SMC controller, which is the reference and which is not described in any way, is there almost 100% overshoot?
The lack of current signal recording makes it impossible to analyse the experimental results.
What is the range of variation of the moment of inertia? For a range in the order of 1:3 - 1:4, a well-chosen PI controller is fully sufficient, and a correctly chosen SMC provides full robustness.
Reviewer 3 Report
This paper presents an integral time-varying fast terminal sliding mode for anti-inertia disturbance speed control of permanent magnet synchronous motor. Some comments are listed as follows.
1. The contribution of this paper is unclear. Various methodology for anti-inertia disturbance speed control of permanent magnet synchronous motor are wildly developed in the literature. More comparisons between different schemes with the proposed method should be given. What is the advantage of the proposed method? The main contributions and advantages should be stressed in detail in this paper. Moreover, the presentation is unclear.
2. How does the b_head be obtained as (13)? B_head depends on e_dot? Is it reasonable?
3. It seems that equation (21) cannot be obtained from (17), (18) and (21). More details are required.
4. In line 265, what is c2?
5. In the experiment, how does the design parameters of the proposed controller be selected? More details are required.
Author Response
Response to Reviewers’ Comments for Reviewer #3
ISSN 2075-1702: Anti-inertia Disturbance Speed Control of Permanent Magnet Synchronous Motor Based on Integral Time-varying Fast Terminal Sliding Mode
The authors would like to thank the reviewers for very constructive and detailed comments that have helped us improve the paper. All the comments and suggestions have been carefully considered. All changes to the manuscript have been highlighted with the BLUE color. The following summarizes our revisions and explanations itemized according to each reviewer’s comments. Please note unless otherwise specified, all the page numbers cited below are for the revised manuscript. Especially, please check the text in BLUE color in the mentioned pages in the following responses.
Reviewer’s Comment:
1.The contribution of this paper is unclear. Various methodology for anti-inertia disturbance speed control of permanent magnet synchronous motor are wildly developed in the literature. More comparisons between different schemes with the proposed method should be given. What is the advantage of the proposed method? The main contributions and advantages should be stressed in detail in this paper. Moreover, the presentation is unclear.
Authors’ Response:
Thanks for the reviewer’s valuable comments.The modified part is as follows:
When the moment of inertia changes, the integral time-varying module of the sliding mode controller designed in this paper can improve the stability of the motor rotational speed, and the overshoot of the rotational speed can be maintained within a minimum error range; The fast terminal module can improve the fast performance of the motor rotation speed and restore stability in a relatively short time.(Page 2,line 46--Page 2, line 51)
As shown in Figure 9, after the introduction of the fast terminal module in ITSMC, the response time of the motor during startup is significantly reduced, and there is almost no overshoot. Compared to the other three sliding mode controllers, ITFTSMC provides a faster and more stable response speed during motor startup. When the moment of inertia changes, both SMC and ISMC have larger overshoots and longer response times, while both ITSMC and ITFTSMC have smaller response times and almost no overshoots. Overall, the rotational speed of the ITFTSMC has the best fast and stable performance.(Page 9,line 13--Page 9, line 14),(Page 10,line 1--Page 10,line 6)
Obviously, the speed stability and fast performance of ITFTSMC are superior to the other three controllers during motor startup.(Page 12,line 15--Page 12,line 18)
5.3.2.1. Effect of small inertia disturbance on static speed performance
To verify the influence of small inertia disturbance on the static speed performance, keep the inertia at a given speed unchanged, and the moment of inertia increases from toat 2s, and the speed response waveform at this time was then compared. Figure 13 shows the rotational speed waveforms of the four sets of methods under low, medium, and high-speed conditions.
Figure 13. Static rotational speed response waveform under small inertia disturbance.
As can be seen from Figure 13, when the rotational inertia increases from to, the rotational speed of a conventional SMC changes most significantly, with the highest overshoot and the longest response time; ITFTSMC has the smallest speed fluctuation and the shortest response time. Under low, medium, and high speed conditions, the speed fluctuation of conventional SMC is about 20 rpm higher than that of ITFTSMC, and the response time is about 0.1 s slower; The speed fluctuation of TSMC is about 4 rpm higher than that of ITFTSMC, and the response time is about 15 ms slower; The rotational speed response time of ITSMC is similar to that of ITFTSMC, but the overall fluctuation of its rotational speed is greater than that of ITFTSMC. Experiments have proven that when the inertia changes in a small range, the rotational speed of ITFTSMC has the best stability and fast performance at both low and medium and high speeds.(Page 12,line 20),(Page 13,line 1--Page 13,line 18)
According to Figure 14, when the moment of inertia increases fromto , the SMC speed change remains the largest and the ITFTSMC speed change remains the smallest. Under low, medium, and high speed conditions, SMC speed overshoot is about 15% higher than ITFTSMC, and the response time is about 50ms slower; TSMC overshoot is about 5% higher than ITFTSMC, and the response time is about 20ms slower; The rotational speed fluctuation caused by ITSMC before and after the inertia mutation is greater than that of ITFTSMC, and the rotational speed fluctuation at high speed is the most obvious. Experiments have shown that when inertia changes in a medium range, its rotational speed still has the best stability and fast performance.(Page 14,line 4--Page 14,line 13)
According to Figure 15, when the moment of inertia increases fromto , the rotational speed of the SMC changes most acutely. Under low, medium, and high speed conditions, its rotational speed increases by about 90 rpm, and the response time also exceeds 0.5s; The rotation speed fluctuation of ISMC is relatively small, but the rotation speed increases by about 20 rpm; The response time also reached about 0.1s; The rotational speed fluctuations of ITSMC and ITFTSMC are minimal, but considering that the rotational speed fluctuations of ITSMC before the inertia change are greater than the rotational speed fluctuations of ITFTSMC, it can be concluded that when the inertia changes over a wide range, the rotational speed of ITFTSMC still has the best stability and fast performance.(Page 15,line 2--Page 15,line 11)
5.3.3.2. Influence of moderate inertia disturbance on dynamic rotational speed performance
To verify the influence of the moderate inertia volume disturbance on the dynamic rotation speed performance, the moment of inertia increases from toat 2s. The experimental rotational speed response waveform is shown in Figure 17 below.
Figure 17. Dynamic speed response waveform under moderate inertia disturbance.
According to Figure 17, when the system injects moderate inertial interference when the speed changes, SMC has the worst rotational speed response performance, the longest response time, and the highest overshoot. Its performance also decreases as the amplitude of rotational speed change increases; The TSMC also has a longer rotational speed response time, a higher overshoot rate, and non-stop oscillation, especially the greater the amplitude of rotational speed change, the longer the rotational speed oscillations take, and the greater the amplitude of oscillations; The response time of ITSMC is about 0.05s, while the response time of ITFTSMC is about 0.02s. The response time of ITSMC is slower than that of ITFTSMC. Obviously, when the system introduces moderate inertial disturbances during speed changes, the ITFTSMC used in this article has better anti-interference ability, and its rotational speed has better stability and fast performance.(Page 16,line 12--Page 16,line 16),(Page 17,line 1--Page 17,line 12)
Experiments show that even if large inertia disturbances are introduced when the motor speed changes, the ITFTSMC designed in this paper has the best anti-interference ability, and the stability and fast performance of the speed are also the best.(Page 18,line 6--Page 18,line 8)
When the rotational speed of the motor encounters inertia changes while maintaining a constant value, the rotational speed will also change, and this change will increase with the increase of the inertia change amplitude. The ITFTSMC designed in this article can shorten the response time and weaken the rotational speed fluctuation when the rotational speed changes abruptly.(Page 18,line 19--Page 18,line 21),(Page 19,line 1--Page 19,line 3)
When the rotational speed of a motor encounters an inertia change when it changes, the amplitude of the rotational speed change is greater than when it encounters an inertia change when the rotational speed remains constant. Especially when the amplitude of the rotational speed change is large, the conventional sliding mode controller has lost its function. However, the ITFTSMC designed in this paper can still stabilize the rotational speed of the motor in a relatively short time.(Page 19,line 4--Page 19,line 10)
- How does the b_head be obtained as (13)? B_head depends on e_dot? Is it reasonable?
Authors’ Response:
The authors are grateful to the reviewer for the constructive comments that have helped us improve the paper.
The self-adapting law is derived as follows:
The adaptive system is represented as a nonlinear feedback regulating system, which is mainly divided into a linear constant forward loop part and a nonlinear feedback loop part. For a nonlinear feedback regulating system, the necessary and sufficient conditions for satisfying Popov hyperstability theory are:
1.The transfer function H (s) of a forward linear loop is a strictly positive real matrix.
2.Inverse nonlinear loop satisfies Popov inequality:.
The schematic diagram of the feedback regulation system in this article is as follows:
For condition 1:, take the linear compensator , and the resulting transfer function is strictly positive real. At this time, there is .
For condition 2:Combining Formula and Formula , it can be obtained:
According to ,, it can be concluded that .
According to , it can be concluded that .
The self-adapting law can be obtained by combining conditions 1 and 2:
The modified part is as follows:
To avoid unnecessary parameter coupling between the moment of inertia and the load torque during inertia identification, the load torque carried by the motor is set to a constant value: . Derivation of Equation (1) yields the following:
(5)
Where the state quantity is defined as, control gain is, control quantity is, and Equation (5) can be expressed as:
(6)
The adjustable model obtained from Equation (6) is as follows:
(7)
Where,,is the estimate of.
If the state tracking error is defined as , then the state error equation is expressed as:
(8)
The variable is also defined as,, where is a linear compensator.
The estimate of the control gain can be expressed by the proportional integral adaptive law:
(9)
Substituting Equation (9) into Equation (8) is obtained:
(10)
where,are nonlinear function of and .
Based on the above conditions, the nonlinear feedback regulation system in this paper can be established as shown in Figure 2.
Figure 2. Schematic diagram of feedback regulation system
The self nonlinear feedback regulating system is mainly divided into a linear constant forward loop part and a nonlinear feedback loop part. For a nonlinear feedback regulating system, the necessary and sufficient conditions for satisfying the Popov hyperstability theory are:
1.The transfer function H (s) of a forward linear loop is a strictly positive real matrix.
2.Inverse nonlinear loop satisfies Popov inequality:
(11)
As can be seen from Figure 2, the transfer function of the forward linear circuit is a strictly positive real matrix.
Take to obtain a strictly positive real matrix , and thus obtain .
For Condition 2: Combining the formula 10 with the formula 11, we can obtain:
(12)
where are finite normal numbers.
According to,, it can be concluded
that.
According to ,it can be concluded that.
Combined with , it can be seen that
(13)
Where bothare the integration coefficient and the scale factor.
Substitute Equation 13 into Equation 9 to obtain:
(Page 4,line 16--Page 5,line 22)
- It seems that equation (21) cannot be obtained from (17), (18) and (21). More details are required.
Authors’ Response:
Thanks for the reviewer’s valuable comments. I'm sorry for my error. Here is the derivation process I have modified:
Define the status variables of the PMSM system as:
(17)
Among them,is the motor reference mechanical angular velocity, is usually a constant, and is the actual mechanical angular velocity. This is obtained according to Equations (2) and (17):
(18)
Where is the torque constant,.
The integral time-varying fast terminal sliding mode surface of the system was then defined as:
(19)
Whereare constant, are Positive odd number, and,,, .
Derivation of formula 19 can be obtained:
(20)
To achieve the global robustness of sliding mode control, the system trajectory needs to be on the time-varying sliding mode surface at the initial moment, so that s=0 yields:
(21)
A new convergence law was defined as shown in Equation (20):
(22)
Where,and is the symbolic function.
From Equations (18), (20), and (22), it can be obtained that the q-axis current is as follows:
(23)
By substituting the rotational inertia identification valueinto Equation 23, the self-adapting law of the identification system can be obtained as follows:
(24)
(Page 7,line 21--Page 8,line 10)
- In line 265, what is c2?
Authors’ Response:
Thanks for the reviewer’s valuable comments.I'm sorry that there was an error in the stability analysis due to an error in the derivation of the q-axis current.is a parameter that should not appear. The stability analysis is as follows:
To verify the stability of the integral time-varying fast terminal sliding mode controller, the Lyapunov function was selected, which is expressed:.
Derivation to this function yields .
Substituting Equation (18) and Equation (20) into the above Equation can be obtained:
Substitute Equation (23) into Equation (25) to obtain:
Obviously, the sliding mode controller designed in this paper satisfies the Lyapunov stability theorem:. The system is asymptotically stable.
(Page 10,line 8--Page 10,line 17)
5.In the experiment, how does the design parameters of the proposed controller be selected? More details are required.
Authors’ Response:
Thanks for the reviewer’s valuable comments.
Parameter selection of sliding surface: :
is an integral constant and has .
In order to shorten or eliminate the arrival stage, a time-varying termis introduced on the sliding surface.is an expression of attenuation, which initially guarantees a larger value to reach the sliding surface at a faster speed, and as time increases, tends to zero to ensure system stability.are constants, and, the larger, the faster the convergence speed of. In order to achieve global robustness of sliding mode control, the trajectory of the system needs to stay on the time-varying sliding mode surface at the initial moment. Therefore, if , we can obtain: .
is the terminal sliding mode constant and has.
To avoid singularity in the terminal sliding mode controller, makepositive odd numbers and .
Parameter selection of approach law :.
Each parameter of the reaching law is constant and has,,.
The modified part is as follows:
The integral time-varying fast terminal sliding mode surface of the system was then defined as:
(19)
Whereare constant, are Positive odd number, and,,, .(Page 7,line 23--Page 7,line 24)
To achieve the global robustness of sliding mode control, the system trajectory needs to be on the time-varying sliding mode surface at the initial moment, so that s=0 yields:
(21)
(Page 7,line 27--Page 7,line 29)
A new convergence law was defined as shown in Equation (22):
(22)
Where,and is the symbolic function.
(Page 8,line 4)

Round 2
Reviewer 2 Report
1. I stand by my opinion that the phrase "Anti-inertia Disturbance Speed Control" is incorrect. It is not the role of the reviewer to determine the titles of the work. However, I point out that the term "robustness" or "insensitivity" (which, however, differ in meaning!) is usually used in the literature of the subject.
2. I accept the improvement, although I still completely do not understand the point of putting the symbols N_r and \omega_m at exactly the same size. It only makes the perception of the work more difficult. At the same time, in the diagrams there is another designation "n"
3. Unfortunately, it is not clear what Figure 1 shows. Was the test performed in a closed speed control system or in an open system. The reduction in velocity ripple is an obvious result of the increase in the moment of inertia (although I don't know how to make this step on the test bench in Figure 10) while the change in the average value of the velocity needs to be explained!
4. Unfortunately, I still don't know what a "Landau observer" is. It is not cited by the Authors, and a check on Google returns 0 results.
Assuming a load torque of 0 is possible during theoretical and simulation analyses, not in the real system (what about friction?).
5. Unfortunately, I still think that the presented derivation of the J estimator is completely unclear. The elaboration ends up giving a PI-type formula, for which there is no given rule for selecting the coefficients k_P, k_I. On top of that, there is a completely arbitrarily selected low-pass filter. So the whole procedure cannot be used by readers.
6. sgn() is not a symbolic function (???) but a sign function!
7. My experience with designing various mechanical systems with the SMC absolutely does not confirm such waveforms as in Figure 7. The lack of even a brief display of how the SMC was made makes this picture unverifiable. The problem of the SMC is the chaterring effect of the control signal, solved in various ways. In the explanation, the authors propose to integrate this signal - which leads to caricatured effects. However, in no case can this be generalized!
8. The results are presented chaotically. I suggest that after initially showing that the ITFTSMC controller is the best, show what its properties are for variable load torque and for variable moment of inertia.
9. Comparing which regulator is "faster" when no numerical parameters of the comparable regulators are given is pointless.
Remark.
A well-designed cascade system of PI controllers will operate without major problems in the range of moment of inertia variation of 1:10. The waveforms will be stable, and the use of a setpoint signal shaping system will ensure robustness.
Reviewer 3 Report
The authors have been answered the reviewer's queries. I have no further comments.